# Realization of higher-order topological lattices on a quantum computer

Jin Ming Koh [1,2,5], Tommy Tai [3,4,5] & Ching Hua Lee [4] ✉

Programmable quantum simulators may one day outperform classical computers at certain tasks. But at present, the range of viable applications with noisy intermediate-scale quantum (NISQ) devices remains limited by gate errors and the number of high-quality qubits. Here, we develop an approach that places digital NISQ hardware as a versatile platform for simulating multi-dimensional condensed matter systems. Our method encodes a high-dimensional lattice in terms of many-body interactions on a reduced-dimension model, thereby taking full advantage of the exponentially large Hilbert space of the host quantum system. With circuit optimization and error mitigation techniques, we measured on IBM superconducting quantum processors the topological state dynamics and protected mid-gap spectra of higher-order topological lattices, in up to four dimensions, with high accuracy. Our projected resource requirements scale favorably with system size and lattice dimensionality compared to classical computation, suggesting a possible route to useful quantum advantage in the longer term.

Recent years have witnessed tremendous progress in the development of programmable quantum simulator platforms, including superconducting transmon- and fluxonium-based processors[1,2], trapped ion systems[3], ultracold atomic lattices[4], and Rydberg atom arrays[5,6]. Numerous successful demonstrations in such platforms have been reported, notably in quantum chemistry[7–9] and quantum many-body dynamics[10–14] contexts. Underpinning the present monumental research effort is the hope that quantum-computational platforms can outperform conventional classical counterparts in a range of useful applications, thereby enabling novel capabilities beyond our current reach. While breakthroughs in quantum advantage have been reported in simulational tasks, existing studies focus on highly specific tailored problems, in particular, random quantum circuit sampling[15] and boson sampling[16], which limit their broader applicability. Finding applications for which quantum platforms provide unique advantages, and pushing the capabilities of near-term noisy intermediate-scale quantum (NISQ) hardware, remain pertinent and timely objectives.

In this work, we develop a new approach that establishes NISQ hardware as a particularly suitable platform for the simulation of generic multi-dimensional condensed-matter lattice models. As a demonstration, we utilize our method to realize, on transmon-based superconducting quantum devices, higher-order topological (HOT) phases in high-dimensional lattices of unprecedented size and complexity. Unlike previous quantum simulator studies that implemented topological models through synthetic dimensions[17–19], we realize HOT lattices in real space, in up to $d = 4$ dimensions (tesseract). Central to our approach is a mapping procedure that encodes single-particle degrees of freedom of a high-dimensional lattice within the many-body Fock space of an interacting one-dimensional (1D) model. This enables us to take full advantage of the exponentially large Hilbert space innately accessible by a quantum computer, while drastically reducing the number of qubits needed for direct simulation.

Importantly, we remark that classical simulation of high-dimensional HOT lattices is expensive, and the resource complexity of our quantum simulation is favorable over classical numerical methods—e.g., exact diagonalization (ED)—in fully general settings. Although the scalable realization of larger lattice systems requires hardware exceeding present capabilities, our approach presents a

[1]Division of Physics, Mathematics and Astronomy, Caltech, Pasadena, CA 91125, USA. [2]A*STAR Quantum Innovation Centre (Q.InC), Institute of High Performance Computing (IHPC), Agency for Science, Technology and Research (A*STAR), 1 Fusionopolis Way, #16-16 Connexis, Singapore 138632, Republic of Singapore. [3]Department of Physics, MIT, Cambridge, MA 02142, USA. [4]Department of Physics, National University of Singapore, Singapore 117542, Singapore. [5]These authors contributed equally: Jin Ming Koh, Tommy Tai. ✉e-mail: phylch@nus.edu.sg

**(a)**

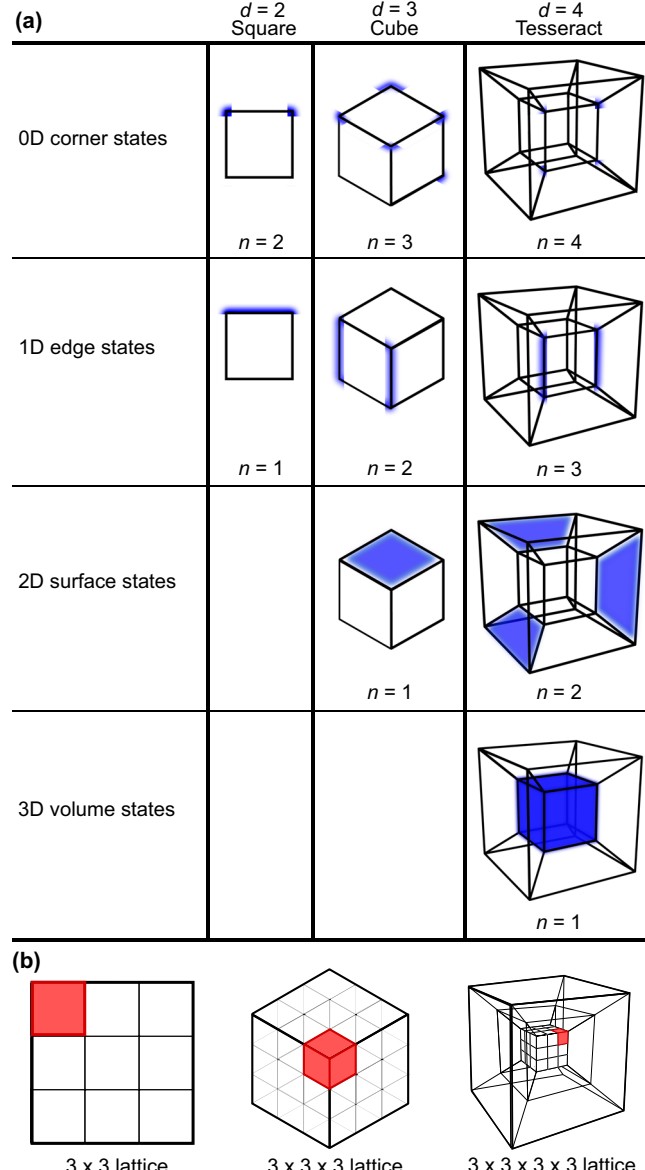

**(b)**

3 × 3 lattice 3 × 3 × 3 lattice 3 × 3 × 3 × 3 lattice

**Fig. 1 | HOT states in higher-dimensional lattices. a** In different number of dimensions $d$, $n$th order topological states manifest as robust codimension-$n$ corner, edge, surface, and volume states protected by HOT invariants. HOT states refer to topological states with $n > 1$. **b** Schematics of HOT corner modes on hyperlattices in different dimensions. The illustrated lattice sizes are small for visual clarity; the HOT states we realized on quantum hardware are on considerably larger $16 \times 16$, $6 \times 6 \times 6$, and $6 \times 6 \times 6 \times 6$ lattices.

possible avenue toward useful quantum advantage as quantum simulator platforms continue to rapidly improve. As the simulation of quantum lattice systems is a ubiquitous foundational task across physical and engineering contexts, such an advantage when realized carries tremendous scientific impact.

Beyond the quantum simulation context, HOT phases of matter are of fundamental interest across condensed matter settings. Underscoring topological insulators[20] and their higher-order counterparts[21–25] is the celebrated bulk-boundary correspondence, which has revolutionized condensed matter physics and provided a fertile setting to realize robust boundary states for promising technological applications[20]. Unlike conventional topological systems, HOT systems support protected modes at higher codimension corners, edges, and surfaces. An $n$th-order HOT system in $d \geq 2$ dimensions hosts

$d_c = (d − n)$-dimensional midgap boundary modes (with $n \leq d$) but is insulating everywhere else, especially the bulk (Fig. 1).

Our unique approach lends a potential re-interpretation of HOT robustness as an interaction-mediated phenomenon between multiple particles in 1D, thereby introducing a new class of 1D many-body systems with topologically protected clustering properties manifesting in the joint configuration space of multiple particles. HOT, in particular, can become an alternative mechanism for enforcing $d$-body clustering or repulsion beyond the scope of known mechanisms such as Coulomb repulsion. Promoting these HOT-protected boundary states to spatial corners further opens avenues for applications, such as the realization of a topological qubit using HOT superconductors that host elusive Majorana corner modes[26]. This complements the search for HOT phases in quantum materials[23,24], which is presently still in its infancy.

## Results and discussion

### Mapping higher-dimensional lattices to 1D quantum chains

While small quasi-1D and 2D systems have been simulated on digital quantum computers[27,28], the explicit simulation of higher-dimensional lattices remains elusive. Directly simulating a $d$-dimensional lattice of width $L$ along each dimension requires $\sim L^d$ qubits. For large dimensionality $d$ or lattice size $L$, this quickly becomes infeasible on NISQ devices, which are significantly limited by the number of usable qubits, qubit connectivity, gate errors, and decoherence times.

To overcome these hardware limitations, we devise an approach to exploit the exponentially large many-body Hilbert space of an interacting qubit chain. The key inspiration is that most local lattice models only access a small portion of the full Hilbert space (particularly non-interacting models and models with symmetries), and an $L^d$-site lattice can be consistently represented with far fewer than $L^d$ qubits. To do so, we introduce an exact mapping that reduces $d$-dimensional lattices to 1D chains hosting $d$-particle interactions, which is naturally simulable on a quantum computer that accesses and operates on the many-body Hilbert space of a register of qubits.

**General mapping formalism.** At a general level, we consider a generic $d$-dimensional $n$-band model $\mathcal{H} = \sum_{\mathbf{k}} \mathbf{c}_{\mathbf{k}}^{\dagger} \mathcal{H}(\mathbf{k}) \mathbf{c}_{\mathbf{k}}$ on an arbitrary lattice. In real space,

$$\mathcal{H} = \sum_{\mathbf{r}\mathbf{r}'} \sum_{\gamma\gamma'} h_{\mathbf{r}\mathbf{r}'}^{\gamma\gamma'} c_{\mathbf{r}\gamma}^{\dagger} c_{\mathbf{r}'\gamma'}, \tag{1}$$

where we have associated the band degrees of freedom to a sublattice structure $\gamma$, and $h_{\mathbf{r}\mathbf{r}'}^{\gamma\gamma'} = 0$ for $|\mathbf{r} - \mathbf{r}'|$ outside the coupling range of the model, i.e., adjacent sites for a nearest-neighbor (NN) model, next-adjacent for next-NN, etc. The operator $c_{\mathbf{r}\gamma}$ annihilates particle excitations on sublattice $\gamma$ of site $\mathbf{r}$.

To take advantage of the degrees of freedom in the many-body Hilbert space, our mapping is defined such that the hopping of a single particle on the original $d$-dimensional lattice from $(\mathbf{r}', \gamma')$ to $(\mathbf{r}, \gamma)$ becomes the simultaneous hopping of $d$ particles, each of a distinct species, from locations $(r'_1, \ldots, r'_d)$ to $(r_1, \ldots, r_d)$ and sublattice $\gamma'$ to $\gamma$ on a 1D interacting chain. Explicitly, this map is given by

$$\mathbf{c}_{\mathbf{r}\gamma}^{\dagger} \mapsto \prod_{\alpha=1}^{d} \left[ \omega_{r_\alpha\gamma}^{\alpha} \right]^{\dagger}, \qquad \mathbf{c}_{\mathbf{r}\gamma} \mapsto \prod_{\alpha=1}^{d} \omega_{r_\alpha\gamma}^{\alpha}, \tag{2}$$

where $r_\alpha$ is the $\alpha$th component of $\mathbf{r}$, and $\{\omega_{\ell\gamma}^{\alpha}\}_{\alpha=1}^{d}$ represents $d$ excitation species hosted on sublattice $\gamma$ of site $\ell$ on the interacting chain, yielding

$$\mathcal{H} \mapsto \mathcal{H}_{1D} = \sum_{\mathbf{r}\mathbf{r}'} \sum_{\gamma\gamma'} h_{\mathbf{r}\mathbf{r}'}^{\gamma\gamma'} \prod_{\alpha=1}^{d} \left[ \omega_{r_\alpha\gamma}^{\alpha} \right]^{\dagger} \omega_{r'_\alpha\gamma'}^{\alpha}. \tag{3}$$

In the single-particle context, exchange statistics is unimportant, and $\{\omega^\alpha\}$ can be taken to be commuting. This mapping framework accommodates any lattice dimension and geometry, and any number of bands or sublattice degrees of freedom. As the mapping is performed at the second-quantized level, any one-body Hamiltonian expressed in second-quantized form can be treated, which encompasses a wide variety of single-body topological phenomena of interest. We refer readers to Supplementary Note 1 for a more expansive technical discussion. With slight modifications, this mapping can also be extended to admit interaction terms in the original $d$-dimensional lattice Hamiltonian, although we do not explore them further in this work.

**Mapping HOT lattices onto 1D qubit chains.** For concreteness, we specialize our Hamiltonian to HOT systems henceforth and shall detail how our mapping enables them to be encoded on quantum processors. The simplest square lattice with HOT corner modes[21] may be constructed from the paradigmatic 1D Su-Schrieffer Heeger (SSH) model[29]. To allow for sufficient degrees of freedom for topological localization, we minimally require a 2D mesh of two different types of SSH chains in each direction, arranged in an alternating fashion

$$\mathcal{H}_{\text{lattice}}^{2D} = \sum_{(x,y)\in[1,L]^2} \left[ u_{xy}^x c_{(x+1)y}^\dagger + u_{yx}^y c_{x(y+1)}^\dagger \right] c_{xy} + \text{h.c.}, \qquad (4)$$

where $c_{xy}$ is the annihilation operator acting on site $(x, y)$ of the lattice and $u_{r_1 r_2}^\alpha$ takes values of either $v_{r_1 r_2}^\alpha$ for intra-cell hopping (odd $r_2$) or $w_{r_1 r_2}^\alpha$ for inter-cell hopping (even $r_2$), $\alpha \in \{x, y\}$. Conceptually, we recognize that the 2D lattice momentum space can be equivalently interpreted as the joint configuration momentum space of two particles, specifically, the $(1+1)$-body sector of a corresponding 1D interacting chain. We map $c_{xy} \mapsto \mu_x v_y$, where $\mu_\ell$ and $v_\ell$ annihilate hardcore bosons of two different species at site $\ell$ on the chain. In the notation of Eq. (2), we identify $\omega_\ell^1 = \omega_\ell^x = \mu_\ell$ and $\omega_\ell^2 = \omega_\ell^y = v_\ell$, and the sublattice structure has been absorbed into the (parity of) spatial coordinates. This yields an effective 1D, two-boson chain described by

$$\mathcal{H}_{\text{chain}}^{2D} = \sum_{x=1}^{L}\sum_{y=1}^{L} \left[ u_{xy}^x \mu_{x+1}^\dagger \mu_x n_y^v + u_{yx}^y v_{y+1}^\dagger v_y n_x^\mu \right] + \text{h.c.}, \qquad (5)$$

where $n_\ell^\omega$ is the number operator for species $\omega$ at site $\ell$ of the chain. As written, each term in $\mathcal{H}_{\text{chain}}^{2D}$ represents an effective SSH model for one particular species $\mu$ or $v$, with the other species not participating in hopping but merely present (hence its number operator). These two-body interactions arising in $\mathcal{H}_{\text{chain}}^{2D}$ appear convoluted, but can be readily accommodated on a quantum computer, taking advantage of the quantum nature of the platform. To realize $\mathcal{H}_{\text{chain}}^{2D}$ on a quantum computer, we utilize 2 qubits to represent each site of the chain, associating the unoccupied, $\mu$-occupied, $v$-occupied and both $\mu$, $v$-occupied boson states to qubit states $|00\rangle$, $|01\rangle$, $|10\rangle$, and $|11\rangle$ respectively. Thus $2L$ qubits are needed for the simulation, a significant reduction from $L^2$ qubits without the mapping, especially for large lattice sizes. We present simulation results on IBM quantum computers for lattice size $L \sim \mathcal{O}(10)$ in the "Two-dimensional HOT square lattice" section.

Our methodology naturally generalizes to higher dimensions. Specifically, a $d$-dimensional HOT lattice maps onto a $d$-species interacting 1D chain, and $d$ qubits are employed to represent each site of the chain, providing sufficient many-body degrees of freedom to encode the $2^d$ occupancy basis states of each site. We write

$$\mathcal{H}_{\text{lattice}}^{dD} = \sum_{\mathbf{r}\in[1,L]^d}\sum_{\alpha=1}^{d} u_{\mathbf{r}}^\alpha c_{\mathbf{r}+\hat{\mathbf{e}}_\alpha}^\dagger c_{\mathbf{r}} + \text{h.c.}, \qquad (6)$$

where $\alpha$ enumerates the directions along which hoppings occur and $\hat{\mathbf{e}}_\alpha$ is the unit vector along $\alpha$. As before, the hopping coefficients alternate between inter- and intra-cell values that can be different in each direction. Compactly, $u_{\mathbf{r}}^\alpha = [1 - \pi(r_\alpha)]v_{\pi(\mathbf{r}_\alpha)}^\alpha + \pi(r_\alpha)w_{\pi(\mathbf{r}_\alpha)}^\alpha$ for parity function $\pi$, intra- and inter-cell hopping coefficients $v_{\pi(\mathbf{r}_\alpha)}^\alpha$ and $w_{\pi(\mathbf{r}_\alpha)}^\alpha$, and $\mathbf{r}_\alpha$ are spatial coordinates in non-$\alpha$ directions—see Supplementary Table 1 for details of the hopping parameter values used in this work. Using $d$ hardcore boson species $\{\omega^\alpha\}$ to represent the $d$ dimensions, we map onto an interacting chain via $c_{\mathbf{r}} \mapsto \prod_{\alpha=1}^{d}\omega_{r_\alpha}^\alpha$, giving

$$\mathcal{H}_{\text{chain}}^{dD} = \sum_{\mathbf{r}\in[1,L]^d}\sum_{\alpha=1}^{d} u_{\mathbf{r}}^\alpha \left[ \left(\omega_{r_\alpha+1}^\alpha\right)^\dagger \omega_{r_\alpha}^\alpha \prod_{\substack{\beta=1 \\ \beta\neq\alpha}}^{d} n_{r_\beta}^\beta \right] + \text{h.c.}, \qquad (7)$$

where $\omega_\ell^\alpha$ annihilates a hardcore boson of species $\alpha$ at site $\ell$ of the chain and $n_\ell^\alpha$ is the number operator of species $\alpha$. In the $d = 2$ square lattice above, we had $\mathbf{r} = (x, y)$ and $\{\omega^\alpha\} = \{\mu, v\}$. The highest dimensional HOT lattice we shall examine is the $d = 4$ tesseract, for which $\mathbf{r} = (x, y, z, w)$ and $\{\omega^\alpha\} = \{\mu, v, \eta, \xi\}$. In total, a $d$-dimensional HOT lattice Hamiltonian has $d \times 2^d$ distinct hopping coefficients, since there are $d$ different lattice directions and $2^{d-1}$ distinct edges along each direction, each comprising two distinct hopping amplitudes for inter- and intra-cell hopping. Appropriately tuning these coefficients allows the manifestation of robust HOT modes along the boundaries (corners, edges, etc.) of the lattices—schematics of the various lattice configurations investigated in our experiments are shown in later sections.

Accordingly, the equivalent interacting 1D chain requires $dL$ qubits to realize, an overwhelming reduction from the $L^d$ otherwise needed in a direct simulation of $\mathcal{H}_{\text{lattice}}^{dD}$ without the mapping. We remark that such a significant compression is possible because HOT is inherently a single-particle phenomenon. See "Methods" for further details and optimizations of our mapping scheme on the HOT lattices considered, and Supplementary Note 1 for an extended general discussion, including examples of other lattices and models.

## Simulation on quantum hardware

With our mapping, a $d$-dimensional HOT lattice $\mathcal{H}_{\text{lattice}}^{dD}$ with $L^d$ sites is mapped onto an interacting 1D chain $\mathcal{H}_{\text{chain}}^{dD}$ with $dL$ number of qubits, which can be feasibly realized on existing NISQ devices for $L \sim \mathcal{O}(10)$ and $d \leq 4$. While the resultant interactions in $\mathcal{H}_{\text{chain}}^{dD}$ are inevitably complicated, below we describe how $\mathcal{H}_{\text{chain}}^{dD}$ can be viably simulated on quantum hardware.

A high-level overview of our general framework for simulating HOT time-evolution is illustrated in Fig. 2. To evolve an initial state $|\psi_0\rangle$, it is necessary to implement the unitary propagator $U(t) = \exp(-i\mathcal{H}_{\text{chain}}^{dD}t)$ as a quantum circuit, such that the circuit yields $|\psi(t)\rangle = U(t)|\psi_0\rangle$ and desired observables can be measured upon termination. A standard method to implement $U(t)$ is Trotterization, which decomposes $\mathcal{H}_{\text{chain}}^{dD}$ in the spin-1/2 basis and splits time-evolution into small steps (see "Methods" for details). However, while straightforward, such an approach yields deep circuits unsuitable for present-generation NISQ hardware. To compress the circuits, we utilize a tensor network-aided recompilation technique[30–33]. We exploit the number-conserving symmetries of $\mathcal{H}_{\text{chain}}^{dD}$ in each boson species, arising from $\mathcal{H}_{\text{lattice}}^{dD}$ and the nature of our mapping (see "Methods"), to enhance circuit construction performance and quality at large circuit breadths (up to 32 qubits). Moreover, to improve data quality amidst hardware noise, we employ a suite of error mitigation techniques, in particular, readout error mitigation (RO) that approximately corrects bit-flip errors during measurement[34], a post-selection (PS) technique that discards results in unphysical Fock-space sectors[30,35], and averaging across machines and qubit chains (see "Methods").

After acting on $|\psi_0\rangle$ by the quantum circuit that effects $U(t)$, terminal computational-basis measurements are performed on the simulation qubits. We retrieve the site-resolved occupancy densities

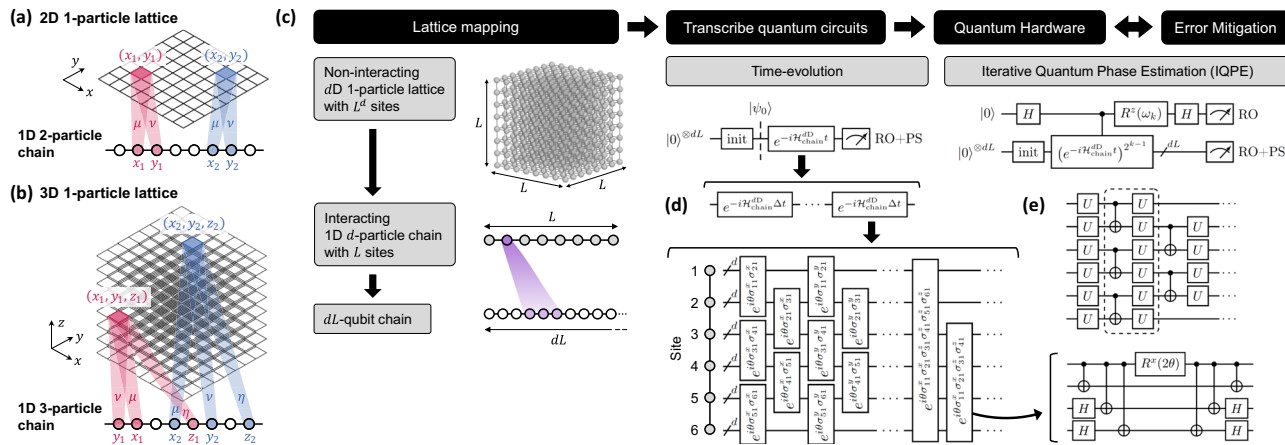

**Fig. 2 | High-level schematic of our approach to simulating high-dimensional lattice models on quantum hardware. a, b** Mapping of a higher-dimensional lattice to a 1D interacting chain to facilitate quantum simulation on near-term devices. Concretely, a two-dimensional single-particle lattice can be represented by a two-species interacting chain; a three-dimensional lattice can be represented by a three-species chain with three-body interactions. **c** Overview of quantum simulation methodology: higher-dimensional lattices are first mapped onto interacting chains, then onto qubits; various techniques, such as **d** Trotterization and **e** ansatz-based recompilation, enable the construction of quantum circuits for dynamical time-evolution, or IQPE for probing the spectrum. The quantum circuits are executed on the quantum processor, and results are post-processed with RO and PS error mitigations to reduce effects of hardware noise. See "Methods" for elaborations on the mapping procedure, and quantum circuit construction and optimization.

$\rho(\mathbf{r}) = \langle c_{\mathbf{r}}^\dagger c_{\mathbf{r}} \rangle = \langle \prod_{\alpha=1}^{d} n_{r_\alpha}^\alpha \rangle$ on the $d$-dimensional lattice, and the extent of evolution of $|\psi(t)\rangle$ away from $|\psi_0\rangle$, whose occupancy densities are $\rho_0(\mathbf{r})$, is assessed via the occupancy fidelity

$$0 \le \mathcal{F}_\rho = \frac{\left[\sum_{\mathbf{r}} \rho(\mathbf{r})\rho_0(\mathbf{r})\right]^2}{\left[\sum_{\mathbf{r}} \rho(\mathbf{r})^2\right]\left[\sum_{\mathbf{r}} \rho_0(\mathbf{r})^2\right]} \le 1. \quad (8)$$

Compared to the state fidelity $\mathcal{F} = |\langle\psi_0|\psi\rangle|^2$, the occupancy fidelity $\mathcal{F}_\rho$ is considerably more resource-efficient to measure on quantum hardware.

In addition to time evolution, we can also directly probe the energy spectrum of our simulated Hamiltonian $\mathcal{H}_{chain}^{dD}$ through iterative quantum phase estimation (IQPE)[36]—see "Methods". Specifically, to characterize the topology of HOT systems, we use IQPE to probe the existence of midgap HOT modes at exponentially suppressed (effectively zero for $L \gg 1$) energies. In contrast to quantum phase estimation[37,38], IQPE circuits are shallower and require fewer qubits, and are thus preferable for implementation on NISQ hardware. As our interest is in HOT modes, we initiate IQPE with maximally localized boundary states that are easily constructed a priori, which exhibit good overlap (>80% state fidelity) with HOT eigenstates, and examine whether IQPE converges consistently towards zero energy. These states are listed in Supplementary Table 2.

**Two-dimensional HOT square lattice**
As the lowest-dimensional incarnation of HOT lattices, the $d = 2$ staggered square lattice harbors only one type of HOT mode—zero-dimensional corner modes (Fig. 1a). Previously, such HOT corner modes on 2D lattices have been realized in various metamaterials[39,40] and photonic waveguides[41], but not in a purely quantum setting to-date. Our equivalent 1D hardcore boson chain can be interpreted as possessing interaction-induced topology that manifests in the joint configuration space of the $d$ bosons hosted on the many-body chain. Here, the topological localization is mediated not due to physical SSH-like couplings or band polarization but due to the combined exclusion effects from all its interaction terms. We emphasize that our physically realized 1D chain contains highly non-trivial interaction terms involving multiple sites—the illustrative example in Fig. 3f for an $L = 6$ chain already contains a multitude of interactions, even though it is much

smaller than the $L = 10$ and $L = 16$ systems we simulated on quantum hardware. As evident, the $d \times 2^d = 8$ unique types of interactions, corresponding to the 8 different couplings on the lattice, are mostly non-local; but this does not prohibit their implementation on quantum circuits. Indeed, the versatility of digital quantum simulators in realizing effectively arbitrary interactions allows the implementation of complex interacting Hamiltonian terms, and is critical in enabling our quantum device simulations.

In our experiments, we consider three different scenarios: C0, having no topological corner modes; C2, having two corner modes at corners $(x, y) = (1, 1)$ and $(L, 1)$; and C4, having corner modes on all four corners. These scenarios can be obtained by appropriately tuning the eight coupling parameters in the Hamiltonian (Eq. (4))—see Supplementary Table 1 for parameter values[42].

We first show that the correct degeneracy of midgap HOT modes can be measured on each of the configurations C0, C2, and C4 on IBM transmon-based quantum computers, as presented in Fig. 3a. For a start, we used a 20-qubit chain, which logically encodes a $10 \times 10$ HOT lattice, with an additional ancillary qubit for IQPE readout. The number of topological corner modes in each case is accurately obtained through the degeneracy of midgap states of exponentially suppressed energy (red), as measured through IQPE executed on quantum hardware—see "Methods" for details. That these midgap modes are indeed corner-localized is verified via numerical (classical) diagonalization, as in the insets of Fig. 3a.

Next, we demonstrate highly accurate dynamical state evolution on larger 32-qubit chains on quantum hardware. We time-evolve various initial states on $16 \times 16$ HOT lattices in the C0, C2, and C4 configurations and measure their site-resolved occupancy densities $\rho(x, y)$, up to a final time $t = 0.8$ when fidelity trends become unambiguous. The resultant occupancy fidelity plots (Fig. 3b–e) conform to the expectation that states localized on topological corners survive the longest, and are also in excellent agreement with reference data from ED. For instance, a localized state at the corner $(x_0, y_0) = (1, 1)$ is robust on C2 and C4 lattice configurations (Fig. 3b), whereas one localized on the $(x_0, y_0) = (1, L)$ corner is robust only on the C4 configuration (Fig. 3c). These fidelity decay trends are corroborated with the measured site-resolved occupancy density $\rho(x, y)$: low occupancy fidelity is always accompanied by a diffused $\rho(x, y)$ away from the initial state, whereas strongly localized states have high occupancy fidelity. In general, the heavy overlap

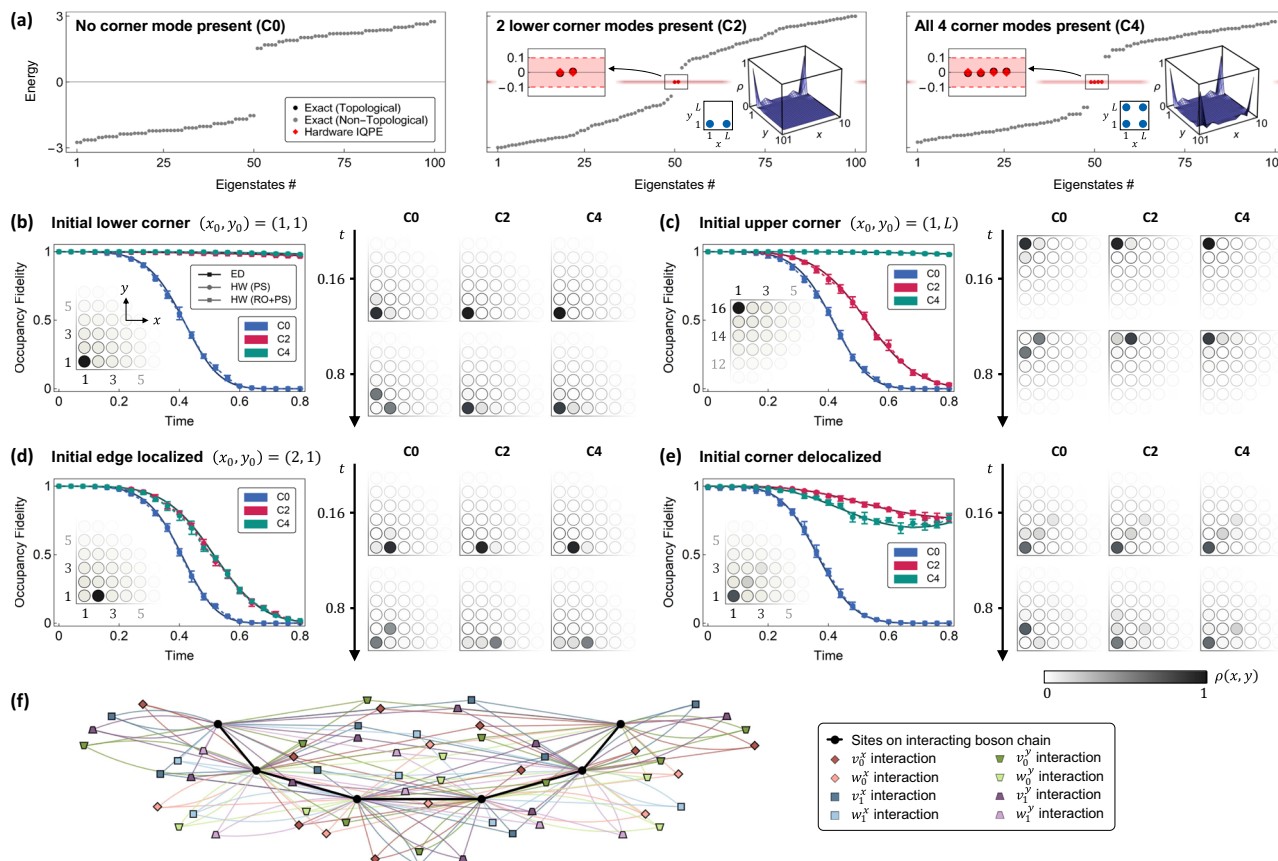

**Fig. 3 | Quantum processor measurements of 2D HOT zero modes and their roles in preserving state fidelity. a** Ordered eigenenergies on a $10 \times 10$ lattice for the topologically trivial C0 and nontrivial C2 and C4 configurations. They correspond to 0, 2, and 4 midgap zero modes (red diamonds), as measured via IQPE on a 20-qubit quantum chain plus an additional ancillary qubit; the shaded red band indicates the IQPE energy resolution. The corner state profiles (right insets) and other eigenenergies (black and gray dots) are numerically obtained via ED. Time-evolution of four initial states on a $16 \times 16$ lattice mapped onto a 32-qubit chain— **b**, **c** localized at corners to highlight topological distinction, **d** localized along an edge, and **e** delocalized in the vicinity of a corner. Left plots show occupancy fidelity for the various lattice configurations, obtained from ED and quantum hardware (labeled HW), with insets showing the site-resolved occupancy density $\rho(x, y)$ of the initial states (darker shading represents higher density). The right grid shows occupancy density measured on hardware at two later times. States with good overlap with robust corners exhibit minimal evolution. Error bars represent standard deviation across repetitions on different qubit chains and devices. In general, the heavy overlap between an initial state and a HOT eigenstate confers topological robustness, resulting in significantly slowed decay. **f** Schematic of the interacting chain Hamiltonian, mapped from the parent 2D lattice, illustrated for a smaller $6 \times 6$ square lattice. The physical sites of the interacting boson chain are colored black, with their many-body interactions represented by colored vertices. Intra- and inter-cell hoppings, mapped onto interactions, are respectively denoted $v_{\boldsymbol{\pi}}^{\alpha}$ and $w_{\boldsymbol{\pi}}^{\alpha}$ for axes $\alpha \in \{x, y\}$ and parities $\boldsymbol{\pi} \in \mathbb{Z}_2^1$.

between an initial state and a HOT eigenstate confers topological robustness, resulting in significantly slowed decay; this is apparent from the occupancy fidelities, which remain near unity over time. In comparison, states that do not enjoy topological protection, such as the $(1, L)$-localized state on the C2 configuration and all initial states on the C0 configuration, rapidly delocalize and decay quickly.

Our experimental runs remain accurate even for initial states that are situated away from the lattice corners, such that they cannot enjoy full topological protection. In Fig. 3d, the initial state at $(x_0, y_0) = (2, 1)$, which neighbors the corner $(1, 1)$, loses its fidelity much sooner than the corner initial state of Fig. 3b, even for the C2 and C4 topological corner configurations. That said, its fidelity evolution still agrees well with ED reference data. In a similar vein, an initial state that is somewhat delocalized at a corner (Fig. 3e) is still conferred a degree of stability when the corner is topological.

**Three-dimensional HOT cubic lattice**
Next, we extend our investigation to the staggered cubic lattice in 3D, which hosts third-order HOT corner modes (Fig. 1a). These elusive corner modes have to date only been realized in classical platforms[43] or in synthetic electronic lattices[44]. Compared to the 2D cases, the

implementation of the 3D HOT lattice (Eq. (6)) as a 1D interacting chain (Eq. (7)) on quantum hardware is more sophisticated. The larger dimensionality of the staggered cubic lattice, in comparison to the square lattice, is reflected by a larger density of multi-site interaction terms on the interacting chain. This is illustrated in Fig. 4b for the minimal $4 \times 4 \times 4$ lattice, where the combination of the various $d = 3$-body interactions gives rise to emergent corner robustness (which appears as up to 3-body boundary clustering as seen on the 1D chain).

On quantum hardware, we implemented 18-qubit chains representing $6 \times 6 \times 6$ cubic lattices in four configurations, specifically, the trivial lattice (C0), two geometrically inequivalent configurations hosting four topological corners (C4a, C4b), and a configuration with all $2^3 = 8$ topological corners (C8). Similar to the 2D HOT lattice, we first present the degeneracy of zero-energy topological modes (header row of Fig. 4a) with low-energy spectral data (red diamonds) accurately obtained via IQPE.

From the first row of Fig. 4a, it is apparent that initial states localized on topological corners enjoy significant robustness. Namely, the measured site-resolved occupancy densities $\rho(x, y, z)$ (four right columns) indicate that the localization of $(x_0, y_0, z_0) = (1, 1, 1)$ corner initial states on C4a, C4b, and C8 configurations are maintained, and

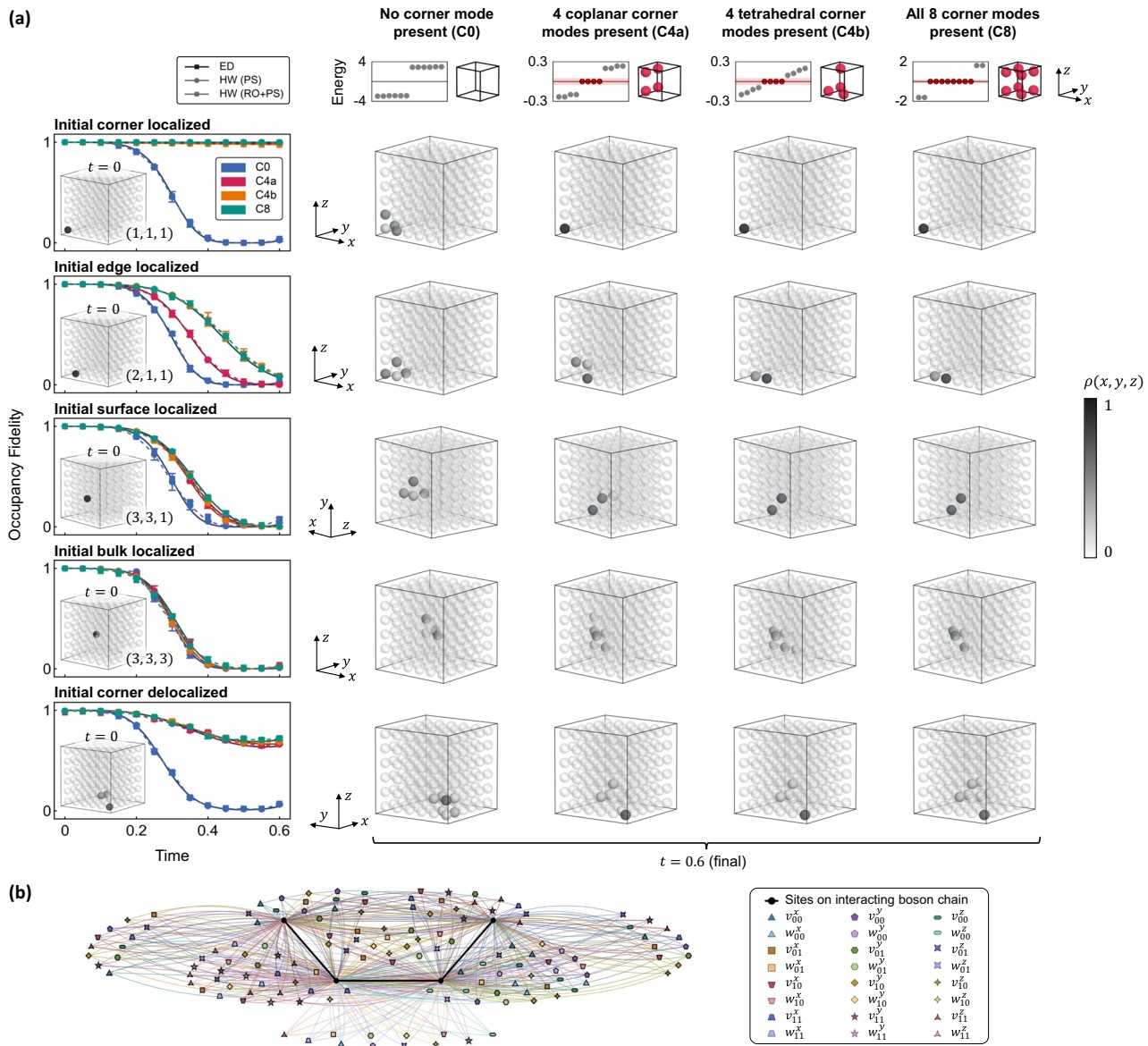

**Fig. 4 | Quantum processor measurements of HOT corner modes on the staggered cubic lattice. a** The header row displays energy spectra for the topologically trivial C0 and inequivalent nontrivial C4a, C4b, and C8 configurations. The configurations host 0, 4, and 8 midgap zero modes (red diamonds), as measured via IQPE on an 18-qubit chain plus an ancillary qubit; the shaded red band indicates the IQPE energy resolution. Schematics illustrating the locations of topologically robust corners are shown on the right. Subsequent rows depict the time-evolution of five initial states on a 6 × 6 × 6 lattice mapped onto an 18-qubit chain—localized at a corner, on an edge, on a face, and in the bulk of the cube, and delocalized in the vicinity of a corner. The leftmost column plots occupancy fidelity for the various lattice configurations, obtained from ED and quantum hardware (labeled HW), with insets showing the site-resolved occupancy density $\rho(x, y, z)$ of the initial state (darker shading represents higher density). The central grid shows occupancy density measured on hardware at a later time ($t = 0.6$), for the corresponding initial state (row) and lattice configuration (column). Error bars represent standard deviation across repetitions on different qubit chains and devices. Again, initial states localized close to topological corners exhibit higher occupational fidelity. **b** Hamiltonian schematic of the interacting chain realizing a minimal 4 × 4 × 4 cubic lattice. Sites on the chain are colored black; colored vertices connecting to multiple sites on the chain denote interaction terms. Intra- and inter-cell hoppings, mapped onto interactions, are respectively denoted $v_{\boldsymbol{\pi}}^{\alpha}$ and $w_{\boldsymbol{\pi}}^{\alpha}$ for axes $\alpha$ $w_{\boldsymbol{\pi}}^{\alpha}$ $\{x, y, z\}$ and parities $\boldsymbol{\pi} \in \mathbb{Z}_2^2$.

measured occupancy fidelities remain near unity. In comparison, an initial corner-localized state on the C0 configuration, which hosts no topological corner modes, delocalizes quickly. Moving away from the corners, an edge-localized state adjacent to a topological corner is conferred slight, but nonetheless present, stability (second row of Fig. 4a), as observed from the slower decay of the $(x_0, y_0, z_0) = (2, 1, 1)$ state on C4a, C4b, and C8 configurations in comparison to the C0 topologically trivial lattice. This conferred robustness is diminished for states localized further from topological corners, for instance, surface-localized states (third row), and is virtually unnoticeable for states localized in the bulk (fourth row), which decay rapidly for all

topological configurations. Initial states that are slightly delocalized near a corner enjoy some protection when the corner is topological, but are unstable when the corner is trivial (fifth row of Fig. 4a). We again highlight the quantitative agreement of our quantum hardware simulation results with theoretical ED predictions.

## Four-dimensional tesseract hyperlattice—HOT corner and edge modes

We now turn to our key results—the NISQ quantum hardware simulation of four-dimensional staggered tesseract HOT lattices. A true 4D lattice is difficult to simulate on most experimental platforms, and with

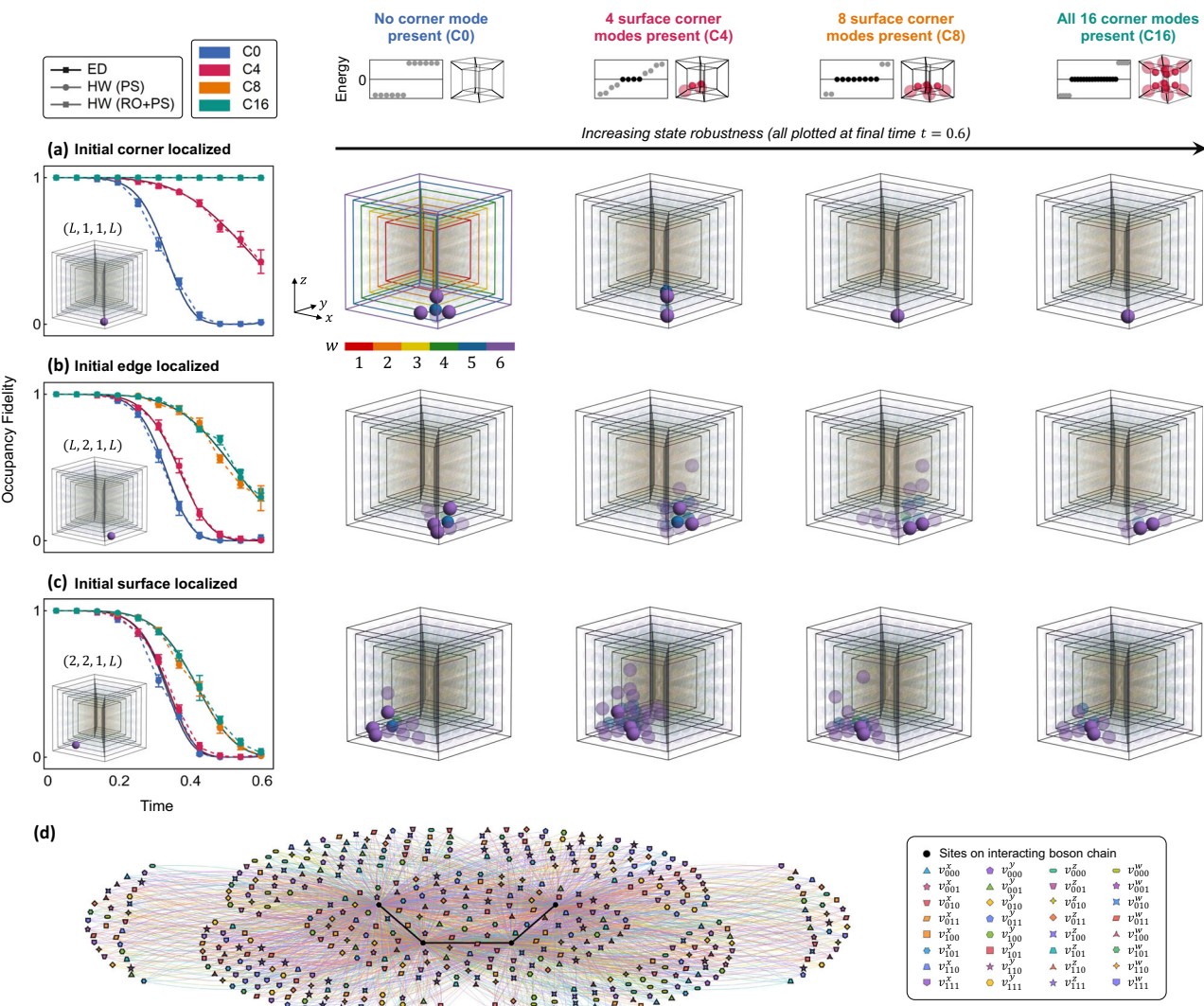

**Fig. 5 | Quantum processor measurements of HOT corner modes on a 4D tesseract lattice.** A $L = 6$ tesseract lattice is illustrated as six cube slices indexed by $w$ and highlighted on a color map. The header row displays energy spectra computed numerically for the topologically trivial C0 and nontrivial C4, C8, and C16 configurations. The configurations host 0, 4, 8, and 16 midgap zero modes (black circles). Schematics on the right illustrate the locations of the topologically robust corners. Subsequent rows depict the time-evolution of three initial states on a $6 \times 6 \times 6 \times 6$ lattice mapped onto a 24-qubit chain—localized on **a** a corner, **b** an edge, and **c** a face. The leftmost column plots occupancy fidelity for the various lattice configurations, obtained from ED and quantum hardware (labeled HW), with insets showing the site-resolved occupancy density $\rho(x, y, z, w)$ of the initial state. Central grid shows occupancy density measured on hardware at the final simulation time ($t = 0.6$), for the corresponding initial state (row) and lattice configuration (column). The color of individual sites (spheres) denotes their $w$-coordinate and color saturation denotes occupancy of the site; unoccupied sites are translucent. Error bars represent standard deviation across repetitions on different qubit chains and devices. Initial states with less overlap with topological corners exhibit slightly lower stability than their lower dimensional counterparts, as these states diffuse into the more spacious 4D configuration space. **d** Hamiltonian schematic of the interacting chain realizing a minimal $4 \times 4 \times 4 \times 4$ tesseract lattice. Sites on the chain are colored black; colored vertices connecting to multiple sites on the chain denote interaction terms. Intra- and inter-cell hoppings, mapped onto interactions, are respectively denoted $v^\alpha_{\boldsymbol{\pi}}$ and $w^\alpha_{\boldsymbol{\pi}}$ for axes $\alpha \in \{x, y, z, w\}$ and parities $\boldsymbol{\pi} \in \mathbb{Z}^3_2$. To limit visual clutter, only $v^\alpha_{\boldsymbol{\pi}}$ intra-cell couplings are shown; a corresponding set of $w^\alpha_{\boldsymbol{\pi}}$ inter-cell couplings are present in the Hamiltonian but have been omitted from the diagram.

a few exceptions[45], most works to date have relied on using synthetic dimensions[18,46]. In comparison, utilizing our exact mapping (Eqs. (6) and (7)) that exploits the exponentially large many-body Hilbert space accessible by a quantum computer, a tesseract lattice can be directly simulated on a physical 1D spin (qubit) chain, with the number of spatial dimensions only limited by the number of qubits. The tesseract unit cell can be visualized as two interlinked three-dimensional cubes (spanned by $x$, $y$, $z$ axes) living in adjacent $w$-slices (Fig. 5). The full tesseract lattice of side length $L$ is then represented as successive cubes with different $w$ coordinates, stacked successively from inside out, with the inner and outer wireframe cubes being $w = 1$ and $w = L$ slices. Being more sophisticated, the 4D HOT lattice features various types of HOT corner, edge, and surface modes (Fig. 1a); we presently

focus on the fourth-order (hexadecapolar) HOT corner modes, as well as the third-order (octopolar) HOT edge modes.

To start, we realized a $dL = 4 \times 6 = 24$-qubit chain on the quantum processor, which encodes a $6 \times 6 \times 6 \times 6$ HOT tesseract. The 4-body (8-operator) interactions now come in $d \times 2^d = 64$ types—half of them are illustrated in Fig. 5d, which depicts only the minimal $L = 4$ case. As discussed in the "Mapping higher-dimensional lattices to 1D quantum chains" section, these interactions are each a product of $d - 1$ density terms and a hopping process, the latter acting on the particle species that encodes the coupling direction on the HOT tesseract. In generic models with non-axially aligned hopping, these interactions could be a product of up to $d$ hopping processes. As we shortly illustrate, despite the complexity of the interactions, the signal-to-

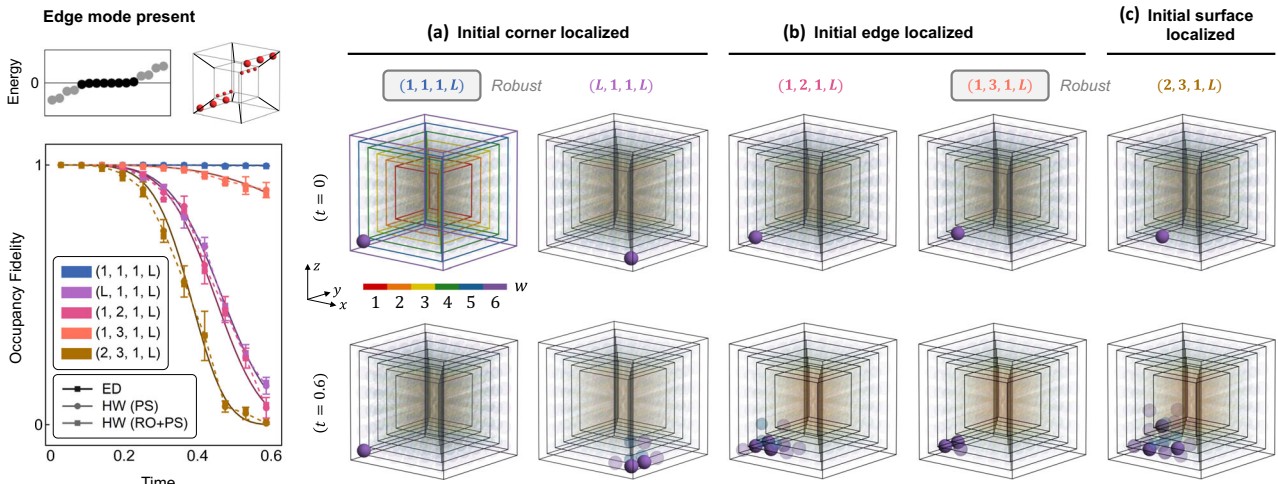

**Fig. 6 | Quantum processor measurements of robustness from HOT edge (or hinge) modes on a 4D tesseract lattice.** Our mapping facilitates the realization of any desired HOT modes, beyond the aforementioned corner mode examples. The header row on the left displays the energy spectrum for a configuration of the tesseract harboring topologically non-trivial edges (midgap mode energies in black). Accompanying schematic highlights alternating sites with topological edge wavefunction support. Subsequent columns present site-resolved occupancy density $\rho(x, y, z, w)$ for a $6 \times 6 \times 6 \times 6$ lattice mapped onto a 24-qubit chain, measured on quantum hardware at $t = 0$ (first row) and final simulation time $t = 0.6$ (second row), for three different experiments. **a** A corner-localized state along a topological edge is robust, compared to one along a non-topological edge. **b** On a topologically non-trivial edge, a state localized on a site with topological wave-function support is robust, compared to one localized on a site without support. **c** A surface-localized state far away from the topological edges diffuses into a large occupancy cloud. The bottom leftmost summarizes occupancy fidelities for the various initial states, obtained from ED and hardware (labeled HW). Error bars represent standard deviation across repetitions on different qubit chains and devices.

noise ratio in our hardware simulations (Fig. 5a) remains reasonably good.

In Fig. 5, we consider the configurations C0, C4, C8, and C16, which correspond respectively to the topologically trivial scenario and lattice configurations hosting four, eight, and all sixteen HOT corner modes, as schematically sketched in the header row. Similar to the 2D and 3D HOT lattices, site-resolved occupancy density $\rho(x, y, z, w)$ and occupancy fidelities measured on quantum hardware reveal strong robustness for initial states localized at topological corners, as illustrated by the strongly localized final states in the C4, C8, and C16 cases (Fig. 5a). However, their stability is now slightly lower, partly due to the more spacious 4D configuration space into which the state can diffuse, as seen from the colored clouds of partly occupied sites after time evolution. Evidently, the stability diminishes as we proceed to the edge- and surface-localized initial states (Fig. 5b and c).

Next, we investigate a lattice configuration that supports HOT edge modes (or commonly referred to as topological hinge states in literature[22]). So far we have seen topological robustness only from topological corner sites (Fig. 5); but with appropriate parameter tuning (see Supplementary Table 1), topological modes can be made to lie along entire edges. This is illustrated in the header row of Fig. 6, where topological modes lie along the $y$-edges. As our HOT lattices are constructed from a mesh of alternating SSH chains, we expect the topological edges to have wavefunction support (nonzero occupancy) only on alternate sites, consistent with the cumulative occupancy densities of the midgap zero-energy modes. This is corroborated by site-resolved occupancy densities and occupancy fidelities measured on quantum hardware, which demonstrate that initial states localized on sites with topological wavefunction support are significantly more robust (Fig. 6a, b), i.e., $(x_0, y_0, z_0, w_0) = (1, 3, 1, L)$ overlaps with the topological mode on $(1, y, 1, L), y \in \{1, 3, 5\}$ sites and is hence robust, but $(1, 2, 1, L)$ is not. The stability of the initial state is reduced as we move farther from the corner, as can be seen, for instance, by comparing occupancy fidelities and the size of the final occupancy cloud for $(1, 1, 1, L)$ and $(1, 3, 1, L)$ in Fig. 6a, b, which is expected from the decaying $y$-profile of the topological edge mode. Finally, our measurements verify that surface-localized states do not enjoy topological protection

(Fig. 6c) as they are localized far away from the topological edges. It is noteworthy that such measurements into the interior of the 4D lattice can be made without additional difficulty on our 1D qubit chain, but doing so can present significant challenges on other platforms, even electrical (topolectrical) circuits.

## Resource scaling

Our approach of mapping a $d$-dimensional HOT lattice onto an interacting 1D chain enabled a drastic reduction in the number of qubits required for simulation, and served a pivotal role in enabling the hardware realizations presented in this work. Here, we further illustrate that employing this mapping for simulation on quantum computers can provide a resource advantage over ED on classical computers, particularly at large lattice dimensionality $d$ or linear size $L$. For this discussion, we largely leave aside tensor network methods, as their advantage over ED is unclear in the generic setting of lattice dimensionality $d > 1$, with arbitrary initial states and evolution time (which may generate large entanglement).

To be concrete, we consider simulation tasks of the following broad type: given an initial state $|\psi_0\rangle$, we wish to perform time-evolution to $|\psi(t)\rangle$, and extract the expectation value of an observable $O$ that is local, that is, $\langle O \rangle$ is dependent on $\mathcal{O}(l^d)$ number of sites on the lattice for a fixed neighborhood of radius $l$ independent of $L$. State preparation or initialization resources for $|\psi_0\rangle$ are excluded from our considerations, as there can be significant variations in costs depending on the choice of specification of the state for both classical and quantum methods. Measurement costs for computing $\langle O \rangle$, however, are considered. To ensure a meaningful comparison, we assume first-order Pauli-basis Trotterization for the construction of quantum circuits, such that circuit preparation is algorithmically straightforward given a lattice Hamiltonian. As a baseline, classical ED of a $d$-dimensional, length $L$ system with a single particle generally requires $\mathcal{O}(L^{3d})$ run-time and $\mathcal{O}(L^{2d})$ dense classical storage to complete a task of such a type[47].

A direct implementation of a generic Hamiltonian using our mapping gives $\mathcal{O}(dL^d \cdot 2^d)$ Pauli strings per Trotter step (see "Methods"), where hoppings along each edge of the lattice, extensive in

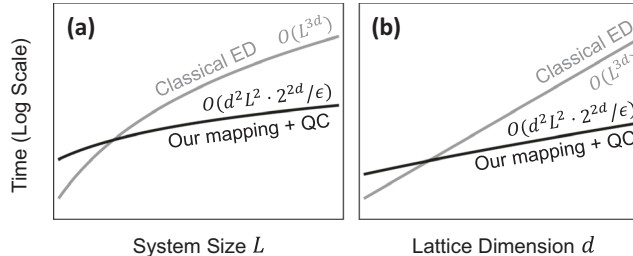

**Fig. 7 | Favorable resource scaling.** Comparison of asymptotic computational time required for the dynamical simulation of $d$-dimensional, size-$L$ lattice Hamiltonians of similar complexity as our HOT lattices. **a** With fixed lattice dimension $d$ and increasing lattice size $L$, the time taken with our approach on a quantum computer (labeled QC) scales with $L^2$, rather than the higher power of $L^{3d}$ through classical ED. **b** For fixed $L$ and varying $d$, our approach scales promisingly, scaling like $4^d$ instead of $(L^3)^d$ for ED. We assume conventional Trotterization for circuit construction, and at large $L$ and $d$, our mapping and quantum simulation approach can provide a resource advantage over classical numerical methods (e.g., ED).

number, are allowed to be independently tuned. However, physically relevant lattices typically host only a systematic subset of hopping processes, described by a sub-extensive number of parameters. In particular, in the HOT lattices we considered, the hopping amplitude $u_{\mathbf{r}}^{\alpha}$ along each axis $\alpha$ is dependent only on $\alpha$ and the parities of coordinates $\mathbf{r}$. Noting the sub-extensive number of distinct hoppings, the lattice Hamiltonian can be written in a more favorable factorized form, yielding $\mathcal{O}(dL \cdot 2^{2d})$ Pauli strings per Trotter step (see "Methods"). Decomposing into a hardware gate set, the total number of gates in a time-evolution circuit scales as $\mathcal{O}(d^2 L^2 \cdot 2^{2d}/\epsilon)$ in the worst-case for simulation precision $\epsilon$, assuming all-to-all connectivity between qubits. Imposing linear NN connectivity on the qubit chain does not alter this bound. Crucially, there is no exponential scaling of $d$ in $L$ (of form $\sim L^d$), unlike classical ED.

For large $L$ and $d$, the circuit preparation and execution time can be lower than the $\mathcal{O}(L^{3d})$ run-time of classical ED. We illustrate this in Fig. 7, which shows a qualitative comparison of run-time scaling between the quantum simulation approach and ED. We have assumed execution time on hardware to scale as the number of gates in the circuit $\mathcal{O}(d^2 L^2 \cdot 2^{2d}/\epsilon)$, which neglects speed-ups afforded by parallelization of single- or two-qubit gates acting on disjoint qubits[48]. The difference in asymptotic complexities implies a crossover at large $L$ or $d$ beyond which quantum simulation exhibits a growing advantage. The exact crossover boundary is sensitive to platform-specific details such as gate times and control capabilities; given the large spread in gate timescales (≳3 orders of magnitude) across present-day platforms[49,50], and uncertain overheads from quantum error correction or mitigation, we avoid giving definite numerical promises on breakeven $L$ and $d$ values. Classical memory usage is similarly bounded during circuit construction, straightforwardly reducible to $\mathcal{O}(dL)$ by constructing and executing gates in a streaming fashion[51], and worst-case $\mathcal{O}(2^{ld})$ during readout to compute $\langle O \rangle$, reducible to a constant supposing basis changes to map components of $O$ onto the computational basis of a fixed number of measured qubits can be implemented on the quantum circuits[52].

The favorable resource scaling (run-time and memory), in combination with the modest $dL$ qubits required, suggests promising scalability of our mapped quantum simulation approach, especially in realizing larger and higher-dimensional HOT lattices. We re-iterate, however, that Trotterized circuits without additional optimization remain largely too deep for present-generation NISQ hardware to execute feasibly. The use of qudit hardware architectures in place of qubits can allow shallower circuits[53]; in particular, using a qudit of local Hilbert space dimension ≥$2^d$ instead of a group of $d$ qubits avoids, to a degree, decomposition of long-range multi-site gates, assuming the ability to efficiently and accurately perform single- and two-qubit

operations[54]. Nonetheless, for the quantum simulation of sophisticated topological lattices as described to be achieved in their full potential, fault-tolerant quantum computation, at the least quantum devices with vastly improved error characteristics and decoherence times, will likely be needed.

## Discussion

Although there exists a plethora of metamaterials and classical systems demonstrating high-dimensional topological or HOT phases[22,39,55–59], these paradigmatic topological states have not been directly simulated on NISQ quantum platforms. Recent demonstrations of various enigmatic condensed-matter phenomena on digital quantum computers, ranging from discrete-time crystals[60] to topological phases[30,61,62], illustrate promising capabilities despite hardware constraints, such as limited gate fidelities and decoherence times. Despite the rapid pace of quantum hardware development and algorithmic advancements, the simulation of HOT phases on quantum computers remains inherently difficult, given their prohibitive qubit number requirements.

By fully exploiting the exponentially large many-body Hilbert space accessible on a quantum computer, we achieved the first simulation of HOT Hamiltonians in up to four dimensions in a fully quantum setting. This is enabled through an exact mapping that encodes the degrees of freedom of the higher-dimensional lattice on a multi-species interacting 1D quantum chain. In the broader context, this mapping is applicable for any lattice geometry and arbitrary configurations of on-site potentials and hoppings on the lattice. On quantum hardware, we not only accurately measured the density evolution of initial states and demonstrated their robustness arising from higher-order topology, but also detected their midgap topological energy spectra, which report on the number of degenerate HOT modes hosted by the high-dimensional lattice. By the renowned bulk-boundary correspondence[63], this is equivalent to probing the corresponding topological invariant of the system. In principle, directly measuring the topological invariant could be achieved via measuring holonomy on wavefunctions[64], but such an approach is costly and challenging to feasibly utilize on generic HOT lattices.

We emphasize that successfully performing a digital quantum simulation of Hamiltonians of comparable complexity to the HOT lattices examined is non-trivial owing to experiment constraints (i.e., qubit numbers, gate errors, and decoherence) on present-day hardware, and the investigation and results presented would have been unviable without the methodology we describe. In the present context, in overcoming the challenges associated with near-term quantum simulation, we pave the way for further experimental investigation of HOT phenomena, valuable especially given the paucity of quantum material HOT candidates. More broadly, our study unveils rich further opportunities in the quantum simulation of higher-dimensional condensed-matter systems harboring novel physics.

Though outside of the present HOT scope, we remark that a similar mapping scheme applies also to interacting lattice, at the expense of a larger local Hilbert space dimension for each quantum chain site and more complicated couplings on the chain (see Supplementary Note 1). Namely, to simulate an interacting $d$-dimensional lattice containing up to $m$ particles, one can employ $m$ sets of the $d$ species of particles hosted on the chain, and in the worst-case, hopping processes on the lattice translate into interactions involving the $md$ particles on the chain. In such a scheme, $\mathcal{O}(mdL)$ qubits suffice to implement the chain, with possible further reductions depending on the lattice. The quantum simulation of interacting $d$-dimensional lattices exhibiting exotic topology is an interesting area for future work.

## Methods
### Quantum hardware
We utilized IBM transmon-based superconducting quantum devices in our experiments. For time-evolution on the square lattice, we used

**(a)**

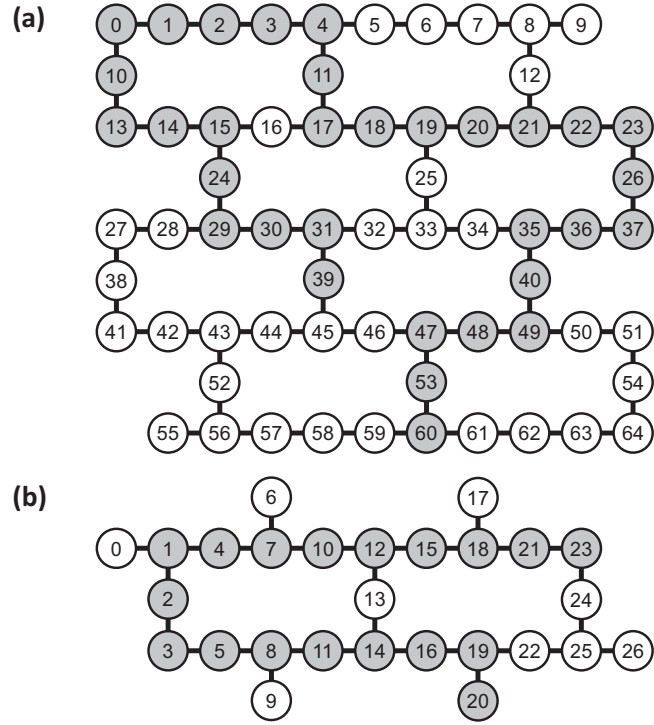

**(b)**

**Fig. 8 | Qubit layout on quantum hardware.** Diagram showing qubit layout and connectivity on **a** 65-qubit and **b** 27-qubit quantum processors. Native CX gates are available on pairs of qubits joined by an edge. An example of a 32-qubit chain in (**a**) and an 18-qubit chain in (**b**), used for 16 × 16 square and 6 × 6 × 6 cubic HOT lattice simulations respectively, are highlighted in gray. The selection of qubit chains is performed via a search scheme that minimizes an estimated error metric (see "Methods").

QV-32 devices *ibmq_manhattan* (65 qubits) and *ibmq_brooklyn* (65 qubits); for time-evolution on the cubic and tesseract lattices, we used QV-32 devices *ibmq_sydney* (27 qubits) and *ibmq_toronto* (27 qubits), and QV-128 devices *ibmq_mumbai* (27 qubits) and *ibmq_montreal* (27 qubits). The latter group of QV-128 devices was also used for IQPE. The quantum volume (QV) reflects an approximate measure of the aggregate capability of the machine—number of qubits, gate error rates, and decoherence times[65]. To provide ballpark measures of performance, the relaxation $T_1$ and dephasing $T_2$ times range 80 μs ≤ $T_1$ ≈ $T_2$ ≤ 130 μs on average on the devices. Typical single-qubit ($\sqrt{X}$) gate errors are $\mathcal{O}(10^{-4})$, and typical two-qubit (CX) gate errors are $\mathcal{O}(10^{-2})$. Illustrations of qubit layouts for 27-qubit and 65-qubit devices are provided in Fig. 8.

**Hardcore bosons**
Our prescribed mapping transforms *d*-dimensional non-interacting lattices to 1D interacting chains hosting hardcore bosons $\{\omega^\alpha\}_{\alpha=1}^d$, which satisfy the standard hardcore bosonic mixed commutation relations

$$\left[\omega_\ell^\alpha, \omega_{\ell'\neq\ell}^\alpha\right] = \left[\omega_\ell^\alpha, \left(\omega_{\ell'\neq\ell}^\alpha\right)^\dagger\right] = \left\{\omega_\ell^\alpha, \omega_\ell^\alpha\right\} = 0, \quad (9)$$

for sites $\ell, \ell'$ on the chain. The anti-commutation relations forbid double occupancy of any site $\ell$ on the chain by two bosons of the same species. We remark that, in the single-particle context of HOT lattices, particle statistics can be neglected since there is no second particle on the *d*-dimensional lattice to exchange with. The choice of hardcore bosons over ordinary bosons in our mapping can be viewed as a form of Fock space truncation supporting simulation on quantum computers; occupancy-dependent phase factors are also avoided compared to fermionic mapping. We may likewise choose $\{\omega^\alpha\}_{\alpha=1}^d$ to be mutually commuting.

**Restricted sector**
The symmetries of our Hamiltonians impose constraints on their physically allowed states. To start, the single-body $\mathcal{H}_{\text{lattice}}^{d\text{D}}$ is manifestly number-conserving. The nature of our mapping in encoding coordinates along axes $\alpha$ as the location of a species-$\alpha$ boson along the chain implies that $\mathcal{H}_{\text{chain}}^{d\text{D}}$ is likewise number-conserving—in particular, exactly one boson of each species occupies the chain at any time. States lying outside this Fock-space sector, which we call the restricted sector, are unphysical. As described later, we take advantage of this constraint in our recompilation and post-selection procedures.

**Representation in terms of qubits**
By definition,

$$\omega_\ell^\alpha = \sum_{\mathbf{m}_\ell \in \{0,1\}^{d-1}} \left|m_\ell^1, \ldots, m_\ell^\alpha = 0, \ldots, m_\ell^d\right\rangle \left\langle m_\ell^1, \ldots, m_\ell^\alpha = 1, \ldots, m_\ell^d\right|, \quad (10)$$

where $|m_\ell^1, \ldots, m_\ell^d\rangle$ are Fock basis states at site $\ell$ on the chain, with $m_\ell^\beta \in \{0,1\}$ indicating the occupancy of species $\beta$. Using $d$ qubits to represent each site of the chain, we associate

$$\left|m_\ell^1, \ldots, m_\ell^d\right\rangle = \bigotimes_{\alpha=1}^d |m_\ell^\alpha\rangle_{\text{q}}, \quad (11)$$

where $|0\rangle_{\text{q}}, |1\rangle_{\text{q}}$ are the computational basis states of a qubit. This fixes the representation

$$\omega_\ell^\alpha = \sum_{\mathbf{m}_\ell \in \{0,1\}^{d-1}} \left[\bigotimes_{\beta=1}^d |m_\ell^\beta\rangle_{\text{q}}\right]_{m_\ell^\alpha = 0} \left[\bigotimes_{\beta=1}^d \langle m_\ell^\beta|_{\text{q}}\right]_{m_\ell^\alpha = 1}$$
$$= \left(\mathbb{I}^{\alpha-1} \otimes \sigma^+ \otimes \mathbb{I}^{d-\alpha}\right)_\ell = \sigma_{\ell\alpha}^+, \quad (12)$$

for Pauli raising and lowering operators $\sigma^\pm = (\sigma^x \pm i\sigma^y)/2$, and $\sigma_{\ell\alpha}^y$ denotes $\sigma^y$ acting on the $\alpha$th qubit of the $\ell$th group, representing the $\ell$th site, each group comprising $d$ qubits. This representation satisfies the mixed hardcore bosonic commutation relations of $\omega^\alpha$ and $(\omega^\alpha)^\dagger$. The number operator

$$n_\ell^\alpha = \left(\omega_\ell^\alpha\right)^\dagger \omega_\ell^\alpha = \sigma_{\ell\alpha}^+ \sigma_{\ell\alpha}^- = \mathbb{I}_{\ell\alpha}^z, \quad (13)$$

for projector $\mathbb{I}_{\ell\alpha}^z = (\mathbb{I}_{\ell\alpha} - \sigma_{\ell\alpha}^z)/2$ onto the $|1\rangle_{\text{q}}$ subspace of the $\alpha$th qubit of the $\ell$th group. Terminal computational-basis ($\sigma^z$) measurements on each qubit on the chain thus suffice to produce site-resolved occupancy density estimates throughout the lattice. Specifically, $\rho(\mathbf{r}) = \langle c_\mathbf{r}^\dagger c_\mathbf{r}\rangle = \langle \prod_{\alpha=1}^d n_{r_\alpha}^\alpha\rangle = \langle \prod_{\alpha=1}^d \mathbb{I}_{r_\alpha\alpha}^z\rangle$ is the probability of detecting $|1\rangle_{\text{q}}$ outcomes on all qubits representing site $\mathbf{r}$ of the lattice. We adopt the convention that a measurement records bit $m \in \{0, 1\}$ for a $|m\rangle_{\text{q}}$ outcome, thus the outcomes for $dL$ qubits are $dL$-length bitstrings. Then, $\rho(\mathbf{r})$ is the fraction of bitstrings with 1s on all qubits $\{r_\alpha\alpha\}_{\alpha=1}^d$. The statistical reliability of $\rho(\mathbf{r})$ improves with the number of measurement samples; in our experiments, we perform ≥32,000 shots for each simulation circuit.

**Trotterization**
Given a generic Hamiltonian $\mathcal{H}$, conventional Trotterization decomposes $\mathcal{H}$ in the spin-1/2 basis and performs time-evolution through a series of small time steps. Writing $\mathcal{H} = \sum_\gamma A_\gamma \sigma^\gamma$ where $A_\gamma \in \mathbb{C}$ and $\sigma^\gamma$ are Pauli strings, the first-order Trotterization scheme is

$$U(t) = e^{-i\mathcal{H}t} = \left(\prod_\gamma e^{-iA_\gamma \sigma^\gamma \Delta t}\right)^M + \mathcal{O}\left(\frac{1}{M}\right), \quad (14)$$

for $M$ time steps (Trotter steps) each covering $\Delta t = t/M$. The error term arises from the non-commutation of $\sigma^\gamma$ strings in $\mathcal{H}$, and implies that a number of steps scaling as $1/\epsilon$ are required to achieve a simulation precision $\epsilon$. Each of the $e^{-iA_\gamma \sigma^\gamma \Delta t}$ terms can be implemented via standard quantum circuit primitives, thus allowing a straightforward transcription of the time-evolution circuit. There exist higher-order formulae of error $\mathcal{O}(1/M^2)$ that produce deeper circuits. In the context of HOT lattices, substituting our representation of hardcore bosonic operators (Eqs. (12) and (13)) into $\mathcal{H}^{dD}_{chain}$ (Eq. (7)) gives the spin-1/2 decomposition

$$\mathcal{H}^{dD}_{chain} = \sum_{\mathbf{r} \in [1,L]^d} \sum_{\alpha=1}^{d} u^\alpha_{\mathbf{r}} \left[ \sigma^-_{(r_\alpha+1)\alpha} \sigma^+_{r_\alpha \alpha} \prod_{\substack{\beta=1 \\ \beta \neq \alpha}}^{d} \mathbb{I}^z_{r_\beta \beta} \right] + \text{h.c.}, \quad (15)$$

so that naively counting, there are $\mathcal{O}(dL^d)$ coupling terms each hosting $\mathcal{O}(2^d)$ Pauli strings, totaling $\mathcal{O}(dL^d \cdot 2^d)$ Pauli strings comprising $\mathcal{H}^{dD}_{chain}$ and thus manifesting in each Trotter step. These Pauli strings have $\mathcal{O}(d)$ weight, that is, they act non-trivially on $\mathcal{O}(d)$ qubits, so their exponentiated forms can each be implemented with $\mathcal{O}(d)$ gates, assuming all-to-all qubit connectivity on hardware.

Imposing NN qubit connectivity, however, necessitates SWAP gates to effect CX operations between non-adjacent qubits; then the maximal distance between qubits acted upon by the Pauli string determines circuit depth. Here, the maximal distance is bounded by the number of qubits $dL$ on the chain, thus each exponentiated Pauli string can be implemented with $\mathcal{O}(dL)$ gates, using a cascading SWAP scheme. Thus, a time-evolution circuit depth of $\mathcal{O}(d^2L^d \cdot 2^d/\epsilon)$ and $\mathcal{O}(d^2L^{d+1} \cdot 2^d/\epsilon)$ are expected for all-to-all and NN qubit connectivities respectively, at simulation precision $\epsilon$.

In the above decomposition (Eq. (15)) and analysis for generic Hamiltonians, we have not assumed any particular structure to the hoppings $u^\alpha_{\mathbf{r}}$; thus the results are applicable to any $d$-dimensional lattice hosting NN hoppings. In our HOT lattices, however, $u^\alpha_{\mathbf{r}}$ depends only on $\alpha$ and the parities of coordinates $\mathbf{r}$, alternating between intra- and inter-cell counterparts. This structure allows a more compact decomposition within the physically relevant sector,

$$\mathcal{H}^{dD}_{chain} = \sum_{\boldsymbol{\pi} \in \{0,1\}^d} \sum_{\alpha=1}^{d} u^\alpha_{\boldsymbol{\pi}} \left[ \sum_{\substack{\ell=1 \\ \pi(\ell)=\pi_\alpha}}^{L} \underbrace{(\omega^\alpha_{\ell+1})^\dagger \omega^\alpha_\ell}_{\sigma^-_{(\ell+1)\alpha}\sigma^+_{\ell\alpha}} \right] \prod_{\substack{\beta=1 \\ \beta \neq \alpha}}^{d} P^\beta_{\pi_\beta} + \text{h.c.},$$

$$P^\beta_\tau = \frac{1}{2} \left[ \mathbb{I}^{\otimes dL} - \prod_{\substack{\ell=1 \\ \pi(\ell)=\tau}}^{L} \sigma^z_{\ell\beta} \right], \quad (16)$$

where $u^\alpha_{\boldsymbol{\pi}}$ is now labeled by the parity vector $\boldsymbol{\pi}$ of lattice coordinates, and the parity operator $P^\beta_\tau |\psi\rangle = |\psi\rangle$ for $|\psi\rangle$ harboring a particle on a parity-$\tau$ position along axis $\beta$ and $P^\beta_\tau |\psi\rangle = 0$ otherwise (parity $1-\tau$). Instead of having $\mathcal{O}(dL^d)$ unique coupling terms, there are $\mathcal{O}(d \cdot 2^d)$ coupling terms each hosting $\mathcal{O}(L \cdot 2^d)$ Pauli strings, thus totaling $\mathcal{O}(dL \cdot 2^{2d})$ Pauli strings comprising $\mathcal{H}^{dD}_{chain}$. The strings are of weight $\mathcal{O}(dL)$, so for both all-to-all and NN qubit connectivity, a time-evolution circuit depth of $\mathcal{O}(d^2L^2 \cdot 2^{2d}/\epsilon)$ is expected. In comparison to the $\mathcal{O}(d^2L^{d+1} \cdot 2^d/\epsilon)$ depth for the naive Trotterization above with NN qubit connectivity, the present scheme turns the factor $L^d \to L \cdot 2^d$, which is significantly advantageous for $L \gg 1$.

### Recompilation

Circuit recompilation starts from a circuit ansatz, whose parameters are dynamically optimized[30–33,66]. We use a circuit ansatz comprising an initial layer of $U_3$ general single-qubit rotation gates on all qubits followed by $K$ ansatz layers, each comprising a layer of CX gates entangling adjacent qubits and a layer of $U_3$ rotations (see Fig. 2e). This ansatz provides sufficient entangling power to accommodate the

couplings in $\mathcal{H}^{dD}_{chain}$ we investigated, at modest $K \leq 10$ depth. Each $U_3$ gate in the ansatz is associated with rotation angles $(\theta, \phi, \lambda)$; we collate all of them into parameter vector $\vartheta$. Then, given a target circuit unitary $V$ and an initial state $|\psi_0\rangle$, the optimization problem

$$\text{argmax}_\vartheta \, \mathcal{F}(V_\vartheta |\psi_0\rangle, V |\psi_0\rangle) = \text{argmax}_\vartheta \left| \langle \psi_0 | V^\dagger_\vartheta V | \psi_0 \rangle \right|^2, \quad (17)$$

can be numerically treated, where $V_\vartheta$ is the circuit ansatz unitary with parameters $\vartheta$. The recompiled circuit is then the ansatz with optimal parameters $\vartheta^*$ fixing the $U_3$ gates. To enhance recompilation performance and the quality of recompiled circuits, we note that we require access only to the restricted sector of $\mathcal{H}^{dD}_{chain}$, as established above; thus we may focus on optimization within this sector. Concretely, we treat

$$\text{argmax}_\vartheta \, \Omega(\mathcal{F}^R(V_\theta |\psi_0\rangle, V |\psi_0\rangle), \Lambda_0), \quad (18)$$

where fidelity $\mathcal{F}^R$ is taken over the restricted sector, such that loss evaluation is made vastly cheaper. Note that PS enforces the symmetries of $\mathcal{H}^{dD}_{chain}$ that forbid access to states outside the restricted sector. A complication is that $\mathcal{F}^R$ no longer has fixed normalization; to stabilize the optimization, we impose a minimum normalization baseline $\Lambda_0$, by regularizing the sector-specific fidelity with a soft-max $\Omega(x,\Lambda) = \Lambda + \log[1 + e^{k(x-\Lambda)}]/\delta$ of reasonable sharpness $\delta$. In our implementation, we estimate $V_\theta$ through auto-differentiable tensor network-based ansatz simulation[67], and use L-BFGS-B with basin-hopping to perform the optimization[68], for $\leq 10^2$ hops and $\leq 10^3$ iterations per hop, terminated at estimated $\mathcal{F}^R \geq 99.9\%$. Recompilation is tuned to never exceed a few minutes per circuit.

### IQPE

Given $U|\psi\rangle = e^{2\pi i \phi} |\psi\rangle$ for a unitary $U$ and an eigenstate $|\psi\rangle$, quantum phase estimation (QPE) measures eigenphase $\phi \in [0, 1)$, in principle to arbitrary precision[37,38]. Setting $U = e^{-i\mathcal{H}t}$ allows the inference of eigenenergy $E = -2\pi\phi/t$ of $|\psi\rangle$, enabling the probing of the spectrum of $\mathcal{H}$. In standard circuit constructions of QPE, each bit of $\phi$ in binary is measured by an ancillary qubit. In comparison, IQPE uses a single ancillary qubit and does not require multi-qubit quantum Fourier transforms[36], and is thus preferable on NISQ devices.

Truncating $\phi = 0.\phi_1\phi_2...\phi_m$ to $m$ bits, the IQPE algorithm iterates $k = m$ to $k = 1$. In the circuit for the $k = m$ iteration, a controlled-$U^{2^{m-1}}$ block is applied, and the ancilla qubit is measured to determine $\phi_m$. The probability $p_0$ of measuring an ancilla state of $|0\rangle_q$ is $\cos^2[(0.\phi_m)\pi] = 1 - \phi_m$ in the absence of noise[36]; amidst noise, or when $|\psi\rangle$ is not an exact eigenstate, the ancilla measurement is no longer deterministic, but a thresholded inference of $\phi_m = 0$ if $p_0 > 1/2$ and $\phi_m = 1$ otherwise can still be applied. Subsequently, in iteration $k$, a controlled-$U^{2^{k-1}}$ block is applied, and a feedback rotation of angle $\omega_k = -2\pi(0.0\phi_{k+1}\phi_{k+2}...\phi_m)$ is applied on the ancilla to remove the phase due to the previous bits, before likewise inferring $\phi_k$. The circuit schematic for iteration $k$ is illustrated in Fig. 2c. The energy resolution of IQPE is determined by the number of iterations $m$ executed.

To probe corner HOT modes, we select $|\psi\rangle$ to be perfectly localized on lattice corners. More precisely, we pick $|\psi\rangle$ to be simple superpositions of corner-localized states, to emulate the generic profiles of topological eigenstates on finite-size lattices. Despite their simplicity, such $|\psi\rangle$ closely resemble exact HOT modes ($\geq 80\%$ fidelities). RO is applied to all qubits, and PS is applied to the simulation qubits to select for the physically relevant restricted sector.

### Readout error mitigation (RO)

RO error refers to the chance of measuring $|1\rangle_q$ when a qubit is in $|0\rangle_q$ or vice versa, which is non-negligible on NISQ devices. A standard method to mitigate these errors is to characterize the measurement bit-flip probabilities of each qubit; then linear inversion can be

performed on raw measurement counts in experiments to recover approximate true measurement counts[34,69,70]. We first describe complete RO. For terminal measurements on $N$ qubits, where $N = dL$ for time-evolution and $N = dL + 1$ for IQPE, the measurement bitstrings that can result are $S = \{0, 1\}^N$. Given a calibration matrix $M$ with an entry $M_{ss'}$ recording the probability of measuring $s \in S$ when the true result should be $s' \in S$, and denoting the raw measurement count $c_s$ and corrected count $c'_s$ for bitstring $s \in S$, we may write

$$c_s = \sum_{s' \in S} M_{ss'} c'_{s'} \iff \mathbf{c} = M\mathbf{c}' \Rightarrow \mathbf{c}' = M^+ \mathbf{c}, \tag{19}$$

where $\mathbf{c} = \begin{bmatrix} c_0 & c_1 & \dots & c_{2^N-1} \end{bmatrix}^\top$ and $M^+$ is the pseudoinverse of $M$. The estimated $\mathbf{c}'$ may carry negative entries due to the approximate $M$; we zero these entries as they are unphysical. A least-squares fit with nonnegative constraints can alternatively be used, but we chose the pseudoinverse to reduce post-processing costs. The matrix $M$ is constructed at the start of each run by running calibration circuits, which prepare the $N$ qubits in state $|s\rangle$ for each $s \in S$ and collect measurement probabilities. Accordingly, $2^N$ calibration circuits are required. This is prohibitively expensive for large $N$, which motivates tensored RO.

In the tensored scheme, we partition the $N$ qubits into $k$ sub-registers, containing $N_1, \dots, N_k$ qubits. Then calibration matrices $M^1, \dots, M^k$ can be acquired separately for each sub-register, and $\mathbf{c}' = (M^1 \otimes \dots \otimes M^k)^+ \mathbf{c}$ likewise applies. In practice, for efficiency, we avoid the explicit tensor product by block-wise operating on $\mathbf{c}$ with the pseudoinverses of the calibration matrices. The scheme requires $2^{\max(N_1,\dots,N_k)} < 2^N$ calibration circuits, since circuits operating on disjoint subsets of qubits can be merged. This reduction in cost, however, is at the expense of neglecting correlations in readout errors between qubits in different sub-registers. In our experiments, we use up to $k = 4$ sub-registers with up to 8 qubits in each.

### Post-selection (PS)
We exploit the symmetry constraints of $\mathcal{H}^{dD}_{\text{chain}}$ to mitigate gate-round errors incurred in our simulation circuits, to a limited extent. Specifically, recall that only the restricted sector of $\mathcal{H}^{dD}_{\text{chain}}$ is physically allowed. Occupancies outside of the sector are unphysical and should not arise during simulation—measurements that detect so thus indicate erroneous results. At no additional cost, we may piggyback on the terminal computational-basis measurements on simulation qubits, which provide site-resolved occupancy densities, to filter and discard such noise-afflicted outcomes[30,31]. Explicitly, we accept counts $c'_s$ of measurement bitstring $s \in S'$,

$$S' = \left\{ s \in \{0, 1\}^N \,\middle|\, \bigwedge_{\alpha=1}^{d} \left( \sum_{\ell=1}^{L} s_{\ell\alpha} = 1 \right) \right\}, \tag{20}$$

where $s_{\ell\alpha}$ denotes the measurement outcome on the $\alpha$th qubit of the $\ell$th group in bitstring $s$. We zero the counts $c'_s$ for $s \notin S'$, which violate symmetries and are unphysical.

### Qubit selection and averaging
The quantum devices we utilize provide more qubits than needed for our experiments, and there are significant variations in the quality of qubits within the same device. We hence use an exhaustive search scheme to select qubits of the lowest estimated error rates to use in our simulations. Given the available CX couplings between qubits, we identify all distinct qubit chains of length $N$, and compute the following fitness function for each chain,

$$Q = 1 - \left\{ \prod_{i=1}^{N} \left[ \left(1 - \varepsilon_{i,i-1}^{\text{CX}}\right)\left(1 - \varepsilon_{i,i+1}^{\text{CX}}\right) \right]^{K/2} \left(1 - \varepsilon_i^{\text{M}}\right) \right\}^{1/N}, \tag{21}$$

where $\varepsilon_i^{\text{M}}$ is the calibrated readout error for qubit $i$, and $\varepsilon_{i,j}^{\text{CX}}$ is the calibrated CX gate error between qubits $i$ and $j$. By convention, $\varepsilon_{1,0}^{\text{CX}} = \varepsilon_{N,N+1}^{\text{CX}} = 0$. The fitness function above is designed to emulate the structure of our recompilation circuit ansatz, with $K \sim 10$ ansatz layers. We pick the qubit chains with lowest $Q$ for our simulations. This selection is only approximate in nature, since the error rates are obtained from the scheduled calibration (~daily) of quantum devices and are subject to drift and fluctuations over time. Illustrations of qubit chains selected through this procedure are shown in Fig. 8 on 65-qubit and 27-qubit processors. Moreover, to average out fluctuations in data and further reduce the effects of hardware noise, we collate error-mitigated measurement bitstring counts over multiple qubit chains per device as selected by our search, and multiple devices (as listed above).

### Parameter selection
The HOT lattice models used can be built from an alternating $d$-dimensional patchwork of alternating SSH chains[29,42]. Each SSH-type chain is characterized by a topological invariant, its winding number. That is, each constituent SSH chain in our lattice is topological when its corresponding hopping strengths satisfy $|v| < |w|$.

All of our HOT corner states are categorized as type-I[42], with support only on a single sublattice and exponentially decaying amplitudes in all orthogonal directions. This is interpreted as a higher-dimensional manifestation of the 1D topology of the SSH model, such that the winding numbers of the edges that intersect a topological corner are all 1. We provide a complete list of hopping parameters $v^\alpha_{\boldsymbol{\pi}}$, $w^\alpha_{\boldsymbol{\pi}}$ for our models in Supplementary Table 1. For a $d$-dimensional lattice with $E$ number of edges, appropriate choices of parameters can be systematically found using simple offsets $\{\delta_j\}_{j=1}^E$ such that the SSH-like coupling parameters satisfy $v = \lambda(1 - \delta)$, $w = \lambda(1 + \delta)$, with $\lambda = -1$ for a subset of edges chosen to generate a $\pi$ flux through each face of the lattice, thereby gapping the spectrum[21]. A positive (negative) choice of $\delta$ for an edge results in it being topologically non-trivial (trivial).

On the 2D lattice, to achieve midgap 1D edge modes, one can tune the hopping parameters away from a configuration with four corner modes (C4) towards one with only two neighboring corner modes. In this transition, two of the corner modes vanish as the bands become gapless along one momentum direction, leading to the emergence of zero-energy 1D edge modes. This approach straightforwardly generalizes to engineer zero-energy (higher-order) edge modes in higher-dimensional lattices, for example on the tesseract lattice.

## Data availability
Data that support the findings of this study are publicly available via the Open Science Framework (https://osf.io/p2v7y/).

## Code availability
Code used in this study is publicly available via Open Science Framework (https://osf.io/p2v7y/).

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

## Acknowledgements

J.M.K. and T.T. thank Wei En Ng and Yong Han Phee of the National University of Singapore for helpful discussions. The authors acknowledge the use of IBM Quantum services for this work. The views expressed are those of the authors, and do not reflect the official policy or position of IBM or the IBM Quantum team. C.H.L. acknowledges the National Research Foundation Singapore grant under its QEP2.0 programme (NRF2021-QEP2-02-P09). This research is supported by the Ministry of Education, Singapore (MOE award number: MOE-T2EP50222-0003).

## Author contributions

C.H.L. initiated and supervised the project, and wrote parts of the manuscript. J.M.K. developed most of the quantum simulation codebase, ran experiments on emulators and hardware, and wrote parts of the manuscript. T.T. contributed to the codebase, ran experiments, and wrote parts of the manuscript. All authors analyzed computational and experiment results. The manuscript reflects the contributions of all authors.

## Competing interests

The authors declare no competing interests.
