## [Peer Review File · Nature Communications]

REVIEWER COMMENTS

Reviewer #1 (Remarks to the Author):

In this paper, the authors propose a method to encode a high-dimensional lattice with non-local many-body interactions as a reduced-dimension model. In this way, higher-order topological states can be simulated on IBM quantum computers. In principle, I cannot agree with the authors that this method can realize quantum advantage with exponentially large Hilbert space. In this sense, I do not recommend to accept this paper for publication in Nature Communications.

For quantum simulation, it is common that the original system will be mapped to a similar but different system, which may demonstrate some key characteristics of the original studied system. The authors showed a method to map higher dimensional lattices to 1D quantum chains. It is known that 1D quantum systems can be generally studied efficiently by many numerical methods, such as MPS and DMRG. To some extent, it means that the entanglement does not satisfy the volume law. So the system can be simulated efficiently with reduction dimension. In this sense, it is not surprise that the method proposed by the authors can achieve similar results. The paper presents the map for several Hamiltonians. In my opinion, those formulations are straightforward.

For NISQ, it is actually difficult to simulate successfully some quantum phenomena mainly due to serious errors and limited number of qubits. Always, some mapping and error mitigation methods will be employed. I think that the method proposed by the authors is not innovative enough. And the results are not significant enough for publication in Nature Communications.

Reviewer #3 (Remarks to the Author):

In manuscript NCOMMS-23-43592-T, the authors proposed to simulate multi-dimensional higher-order topological models on quantum hardware. The key point lies in the sufficient reduction of lattice sizes by mapping single-particle high-dimensional lattices to one-dimensional interacting qubit chains. Compared with previous quantum simulations through using synthetic dimensions, they realized higher-order topological states in real-space high-dimensional lattices. The results are clear and concrete. To the best of our knowledge, it is the first experimental realization of 4D higher-order topological states in quantum systems. However, it is unclear whether they can use their mapping scheme to study more complex structures such as kagome lattice which can also support higher-order corner states [Nature Materials 18, 108 (2019), Nature Photonics 13, 697 (2019)]. If their

scheme is only applicable to simple square, cubic, tesseract lattices, this work may lack broad research interest. Furthermore, even without performing experiments, constructing higher-order topological corner states from one-dimensional SSH-type model is well known [PRL 118, 076803 (2017), PRL 122, 233902 (2019), PRB 104, 224303 (2021)]. These results are quite natural and expected and of less novelty. I think the authors should also carefully consider the following comments/suggestions:

(1)The mapping scheme can reduce the number of qubits from L^d to dL , which significantly save resources of quantum devices. However, for a single-particle problem, the dimension L^d is accessible in classical computer. The quantum advantages are not obvious in dealing with single-particle higher-order topological problem. Actually, to demonstrate corner states, the size $L=10$ is enough. One can immediately obtain $L^4=10000$ eigenstates, eigenvalues and the dynamics on laptop computer.

(2)The authors should comment whether their scheme can apply to general lattice such as triangle lattice and kagome lattice. Besides, since the results of two-dimensional SSH model are well known, the authors should properly cite existing references, and explicitly show how to apply their mapping scheme to a more complex lattice such as kagome lattice or Benalcazar-Bernevig-Hughes (BBH) lattice.

(3)Fig. 7 should give values of both x and y axis, so that we can know what are the transition points when the mapping scheme with quantum computer is better than classical exact diagonalization.

(4)In Sec.II C, the authors declare "Previously, such HOT corner modes on 2D lattices have been realized in various metamaterials [34, 35]...". It should be noted that the topological corner states in two-dimensional lattices have been experimentally observed in photonic waveguides [PRB 105, 195129 (2022)].

(5)In Sec. II C, the authors claimed that "Our equivalent 1D hardcore boson chain is also the first demonstration of interaction-induced higher-order topology...". We do not think it is interaction-induced higher-order topology. The authors used one dimensional interacting boson chain to simulate two dimensional non-interacting higher-order topological lattice. In one dimensional systems, there is no higher-order corner states at all. They can make such claim only when the original two dimensional lattice is trivial and interaction between particles in two dimensions gives rise to the higher-order topological states.

(6) Ranging from two-dimensional square lattice and three-dimensional cubic lattice to four-dimensional tesseract hyperlattice, the direct measurement of dynamical evolution and energy spectrum serves as evidence for higher-order topological states. Not limited to the observation of topological boundary states, it is better to comment how to experimentally measure the topological invariant on quantum hardware. This is important for bulk-boundary correspondence as you mentioned in the fourth paragraph of the introduction.

Reviewer #4 (Remarks to the Author):

The authors present how to map a single particle hopping on a d-dimensional lattice to d different particles hopping simultaneously on a 1-dimensional line. The original single particle creation (annihilation) operator in one certain dimension is replaced by the product of a specific creation (annihilation) operator of one kind with **(d-1)** number operator of other kinds (representing rest dimensions). In other words, in the 1-dimensional line, each site now contains d qubits representing original dimensions.

The authors then demonstrate impressively 2d,3d,4d robust topological corner modes on IBM hardware.

Overall, I can recommend publication after the following issues are addressed.

1. If my understanding is correct (first paragraph of this report), then eq 5 and eq 7, eq15 are not compatible. The non-hopping dimensions represented by $\prod_{\beta=1}^d n_{r_\beta}^\beta$ in eq7 should be replaced with $\prod_{\beta=1, \beta \neq \alpha}^d n_{r_\beta}^\beta$, since there are **d-1** (not d) degrees of freedom left.
2. This new protocol seems plausible, but eq 2,3,5 are given without proof. It is quite abstract and complicated at first sight, could the authors give some justifications? The proof can be intuitive or mathematical.
3. The original model has 1 excitation in total, now the new model introduces d different particles, it's not clear to me how the new model conserves the total excitation.
4. I'm not fully follow the explanation why there are $d \cdot 2^d$ hopping coefficients under eq 7. Now, d kinds of single particles have 2^d total states (occupied/ unoccupied for each species). Each site in the 1 d chain comprises d qubits, the nearest-neighbor hopping is now mapped on to

d qubits on site l interacting with d qubits on site $l+1$. For all possible cases, there should be $2^d \cdot 2^d$ types of interaction. But since each particle has nearest neighbor hopping only with its own kind (hopping only occurs along one specific dimension), and conserves total excitation number, so interaction terms can be further reduced.

5. Although using less qubits, the authors' protocol would require both long range and **many-body interaction** to create the required hopping between certain species. In section F, resources are counted in terms of Pauli-strings not the actual layers of two-qubits gates compiled on the hardware. This is not fair. I was wondering even after some tensor-network optimization how many layers of two-qubit gates used, say simulate the interactions appear in figure 3 (f)?
6. There is a straightforward and well-structured way to map a d dimension lattice onto a 1d chain. For example, we can flatten the 2-d lattice into 1d line(as shown below), the

This figure is taken from PRB 100, 075138 (2019), but the idea has been known for a long time.

horizontal interactions are kept intact, but the vertical interactions now become the long range two-body interaction. I think at least for 2d case, this mapping may yield better result in terms of two-qubit gates than the authors.

7. The results on hardware are quite impressive. I have some doubts about missing details. The total running time in figure 3,4,5 is normalized to 1, but how does it compares to the qubit coherence time ?

8. In the caption of figure 3,4,5, symbols like v_0^x are not defined. The definitions only appear in the very end of the paper.
9. I suggest replace RO (readout Error Mitigation) with EM, it took a while to search for this unfamiliar term.
10. In the supplementary table, it is not clear how the parameters (v_0^x etc) are mapped to the SSH-like model.

Revision Report for Manuscript NCOMMS-23-43592-T

“Realization of Higher-Order Topological Lattices on a Quantum Computer”

Jin Ming Koh, Tommy Tai, Ching Hua Lee

Reviewer # 1

1. **Comment:** *In this paper, the authors propose a method to encode a high-dimensional lattice with non-local many-body interactions as a reduced-dimension model. In this way, higher-order topological states can be simulated on IBM quantum computers. In principle, I cannot agree with the authors that this method can realize quantum advantage with exponentially large Hilbert space. In this sense, I do not recommend to accept this paper for publication in Nature Communications.*

For quantum simulation, it is common that the original system will be mapped to a similar but different system, which may demonstrate some key characteristics of the original studied system. The authors showed a method to map higher dimensional lattices to 1D quantum chains. It is known that 1D quantum systems can be generally studied efficiently by many numerical methods, such as MPS and DMRG. To some extent, it means that the entanglement does not satisfy the volume law. So the system can be simulated efficiently with reduction dimension. In this sense, it is not surprise that the method proposed by the authors can achieve similar results. The paper presents the map for several Hamiltonians. In my opinion, those formulations are straightforward.

Response: We thank the reviewer for sharing their critical perspective. The reviewer compares our work with generic quantum simulation works that involves first a “reduction dimension” via a choice of mapping, followed by established numerical methods to simulate the mapped system, such as MPS and DMRG; with that in mind, the reviewer comments that our results are “not surprising” and disagrees that our method “can realize quantum advantage”. We sincerely apologize that the distinction of our work has been unclear, in comparison to standard approaches and prior art. Below we address in a detailed manner each of the reviewer’s comments, and we hope the explanations we provide allow a clearer understanding of the motivation and novelty of our work, and the impact our study could bring to the community. We have also made significant corresponding edits in the revised manuscript, which we describe at the end of the response.

Mapping.

We first clarify that our choice of mapping is not a standard ‘textbook’ prescription in quantum simulation. Our main scientific motivation is to maximally exploit the quantum nature of the simulation platform in implementing many-body interactions. Our major points of departure from standard methodology in literature (that the reviewer has generically outlined), which we describe below, **is crucial in making the physical realization of higher-order topological states at all possible on current-day quantum hardware.**

Our mapping from d -dimensional HOT lattices to 1D quantum chains is exact and fully describes all independent degrees of freedom, *i.e.* the 1D quantum chain exhibits precisely the characteristics of the HOT lattice. But unlike other well-established exact mapping formulations (*e.g.* Jordan-Wigner, Bravyi-Kiteav, *etc.*), our mapping applied to the higher-dimension lattices we examine uniquely enables us to use a reduced number of qubits in our simulations (from L^d of a straightforward site-qubit identification to dL qubits). These properties of the mapping we develop underpins the investigation of HOT modes in our experiments, wherein site-resolved occupancies of the original d -dimensional lattice were recovered from measurements of the 1D chain, and our measurements of the zero-energy topological spectrum. The expense of our mapping is that single-body terms in the lattice Hamiltonian are turned into many-body interactions on the chain, but implementing these on a quantum platform is not particularly an issue—this is precisely the benefit of using a quantum device. In all facets, we emphasize the critical importance of this carefully designed methodology in enabling the experiments on current hardware.

For additional reference if helpful, we remark that we provide expanded technical discussions on the formalism and general applicability of the mapping in our responses to comment #3 of Reviewer #2 and comment #4 of Reviewer #3. We have also added a new section to the Supplementary Information, Supplementary Note A, dedicated to discussing the mapping, example, and generalizations.

Classical simulation of 1D systems.

The reviewer suggests other methods to simulate 1D quantum systems efficiently, such as MPS and DMRG. We thank the reviewer for this astute comparison, which is indeed an important consideration. As we outlined in detail below, the **structure of our model precludes straightforward efficient simulation guarantees**, and our quantum simulation approach is not limited by the same constraints as in classical numerical methods.

To the best of our knowledge, classical numerical methods such as MPS require 1D quantum systems to possess *fixed local Hilbert space dimension* and be *confined within the area law regime* for efficient simulation. The latter condition is clear, as was also mentioned by the reviewer. The required bond dimension in tensor network methods is dependent on the amount of entanglement in the system; volume-law systems possess extensive entanglement and require exponentially growing bond dimension in the system size. This is not a unique limitation of MPS—more advanced methods such as PEPS and MERA are likewise subject fundamentally to entanglement constraints. The former condition of fixed local Hilbert space dimension subtly accompanies this consideration. In saying “area law” or “volume law”, one considers the system size (*i.e.* number of sites in the system) as the scaling parameter—but the amount of entanglement that a system of L sites can contain is dependent also on the local Hilbert space dimension s of each of the sites, in particular, the maximum number of bits of entanglement entropy goes as $\sim L \log_2 s$. Were one to allow arbitrarily growing local Hilbert space dimension, one can “hide” extensive (or even super-extensive) entanglement in the “area law” regime, thus breaking tensor-network methods.

In our formulation, we clarify that the local Hilbert space of our 1D interacting quantum chain is not fixed—for a d -dimensional HOT lattice we introduce d hardcore boson species on the chain, thus the local dimension of each of the chain sites is 2^d . This follows from the structure of our mapping and is central to its functioning—otherwise one cannot have sufficient degrees of freedom for an exact mapping. Thus, on definitional and structural grounds, it is not generically true that our 1D quantum chain is efficiently simulable by

standard classical methods (*e.g.* MPS as mentioned).

Moreover, we comment that the many-body (d -body) interactions in our chain model, which are of range stretching up to the length of the chain, pose considerable challenges to tensor-network methods, for which the efficient tensor decomposition of degrees of freedom rely inherently on spatial locality. As a concrete example, in MPS methods, in each time step, after the application of each layer of time-evolution operators (*i.e.* trotterized hoppings and interactions) which generically merge (entangle) local tensors with one another, one must decompose the joined tensor back into MPS form via a series of SVDs followed by bond dimension truncation. With long-range many-body interactions of support up to the length of the chain, entanglement is created between each site and almost every other, and the cost of the SVDs scale with both L and d , generically exponentially bad in both. This is in contrast to k -local interactions, say $k = 2$ for nearest neighbour, which only joins up neighboring site tensors and consequently restoring the MPS form is efficient. Finally, we remark that DMRG as mentioned by the reviewer is applicable to ground-state or low-energy physics but is not a method for simulating time-dynamics, and thus is not suitable for our context—but, for example, MPS with TEBD is.

We wish to assure the reviewer that the research in our group spans in large part classical simulation and tensor networks as well, and we would not have pursued the presented line of experiments (which require significant resources) had our model been identifiable as efficiently simulable under standard conventions.

Route towards quantum advantage.

Having concretely discussed the scientific motivation of our mapping scheme, and evaluated the application of numerical methods (such as those suggested by the reviewer) in the context of our problem, we would now like to qualify the quantum advantage our approach provides. We acknowledge the caution with which the reviewer has approached this topic, and we agree that analyses of this type must be done with extensive care.

We emphasize that our statements about resource advantages (*i.e.* computation time) provided by our quantum simulation methodology are in the asymptotic sense (thus a “route” to advantage)—that is, as the lattice linear size L or dimension d are increased. In particular we presented time complexities for the quantum simulation approach, $\mathcal{O}(d^2 L^2 \cdot 2^{2d}/\epsilon)$, versus classical exact diagonalization $\mathcal{O}(L^{3d})$, in Section II F of the main text, with detailed derivations and relevant considerations in Methods, “Trotterization”. The analysis we provide is based on explicit gate counts of the trotterized quantum circuits for time-evolution and is rigorous. As noted earlier, other standard numerical methods either do not accommodate our model or do not provide provable efficiency guarantees; thus we consider exact diagonalization as the fully general classical competitor to the quantum simulation methodology. The **difference in complexities imply that a crossover boundary of L and d exists** beyond which the quantum simulation approach is advantageous in resource requirements. In fact, as we note in our discussion in Section II F, we are likely over-estimating the $\mathcal{O}(d^2 L^2 \cdot 2^{2d}/\epsilon)$ complexity of the quantum simulation approach as we neglect the parallelization of gates acting on disjoint qubits, which is naturally available on most quantum platforms.

Nevertheless, we fully agree with the reviewer that the statement of quantum advantage is subtle and must be treated with caution—indeed a common lesson within the community in recent years. In particular we cannot preclude the development of novel classical algorithms exhibiting drastic performance improvements, possibly fine-tuned for a particular model. On the flip side the development of capabilities on quantum hardware is, too, diffi-

cult to predict—for example the ability to engineer multi-qubit gates, or highly parallelizable fault-tolerant operations. With sufficiently robust quantum hardware, there will come a point when our approach gives rise to a resource advantage; at the current rapid pace of progress, this regime may come within reach in 3–5 years. While we avoid giving definite projections on the crossover point, we remark that we have performed a set of calculations of the rough magnitude of the breakeven L and d under varying assumptions, in our response to comment #5 of Reviewer #3.

In full consideration of the reviewer’s comments, we have edited relevant sections of our manuscript, for example the Abstract and Introduction, to tone down our mentions of quantum advantage; in all remaining cases, such as in Section II F and Methods, we have ensured that we provide the necessary context in our discussion. Section II F continues to provide detailed information of our analysis. We have also removed the explicit mention of “quantum advantage” in the Title of our manuscript to avoid ambiguity—readers would find the analysis with full contextual information in our main text.

In all, we sincerely thank the reviewer for the critical assessment, and we hope we have addressed all the raised concerns.

2. **Comment:** *For NISQ, it is actually difficult to simulate successfully some quantum phenomena mainly due to serious errors and limited number of qubits. Always, some mapping and error mitigation methods will be employed. I think that the method proposed by the authors is not innovative enough. And the results are not significant enough for publication in Nature Communications.*

Response: We thank the reviewer for their critical perspective and astute comments. We agree completely that in the NISQ era, it is “difficult to simulate successfully some quantum phenomena mainly due to serious errors and limited number of qubits”. This difficulty precisely underscores the novelty and value of our work, and our contributions lie not only in concretely demonstrating that the simulation of complex systems such as higher-dimension HOT lattices is possible on current-day hardware—to the best of our knowledge this is the first ever physical realization of this class of systems on digital quantum platforms—but also in the methods that we develop and utilize to overcome the constraints that the reviewer astutely points out.

In fact, as we discuss in the main text (*e.g.* Section II A 2) and have further contextualized in our response to comment #1, our mapping scheme is intended precisely to address constraints posed by the limited number of qubits on NISQ devices. Specifically, we map the higher-dimensional HOT lattice onto a 1D quantum chain as doing so enables a reduction in the number of qubits required for simulation, as compared to a direct site-to-qubit mapping on the HOT lattice. The mapping also lifts us from the direct requirement of higher-dimensional connectivity between qubits on the lattice.

Moreover, the error mitigation methods we employ, such as readout error mitigation and post-selection exploiting the number-conserving symmetries of the Hamiltonian, precisely address gate errors and hardware noise. In particular, readout error mitigation corrects for inaccuracies in measured observables caused by state discrimination error in measurements (on the order of 1–3% per measurement and are non-negligible on current-day devices), and post-selection detects deviations of the evolved quantum state from the physical d -boson subspace mandated by our mapping, that are caused by hardware noise. **The quality of simulation we have achieved *despite* noisy hardware indicates that these efforts**

are fruitful and useful, and arguably significant enough to be considered for publication in a high-impact journal like *Nature Communications*.

The reviewer has comments that our work is “not significant enough” and “not innovative enough”, likely based on the ground that “always (in quantum simulation), some mapping and error mitigation methods will be employed”—but we emphasize that effective development and use of mapping and error mitigation methods, in particular tailored formulations of both that uniquely benefit the problem at hand and enable the frontier of achievable results in quantum simulation to be advanced, **are of scientific and practical value to the community**. We point out that there remains a steady stream of NISQ quantum simulation works published in Nature-family journals in recent years, such as [Nature Communications 15, 211 (2024)], [Nature 618, 500–505 (2023)], and [npj Quantum Information 9, 72 (2023)], strongly indicating that cutting-edge research performed on NISQ hardware (that utilize mapping and error mitigation techniques) is a relevant endeavor recognized by the community.

In this regard, our mapping to encode any high-dimensional lattice to an interacting quantum chain is far from trivial. We reduce the number of sites in the system from L^d in the original lattice to L on the chain, with an accompanying reduction in the number of qubits required from L^d to dL . This is at the cost of introducing different types of d -body interactions on the chain; but we utilize a digital quantum platform to perform quantum simulation, which has the versatility of accessing and composing operations within its many-body Hilbert space, to implement these interactions without particular issue. This is precisely the advantage of using quantum hardware. This maximal leveraging of capabilities of quantum simulators is supported by a recent perspective [Nature Communications 15, 2123 (2024)] which astutely pointed out that “while some many-body interactions are not physically realizable in analog quantum simulators, their digital decomposition can very well be implemented on a gate-based platform”. While proof-of-principle studies have reported digital quantum simulation of simple models, such as 1D or quasi-1D spin chains and fermionic/bosonic tight-binding models, **no-one before us has concretely demonstrated the successful quantum simulation of models of complexity and scale** comparable to our d -dimensional HOT lattices on current hardware, nor proposed a scalable route for sustained advancement of the simulation of this type of models as hardware improves.

Our results carry far-reaching implications because, foundationally, we utilize the “quantumness” of quantum hardware to our full advantage—namely the many-body Hilbert space and many-body interactions the platform supports. As quantum hardware improves in accuracy and circuit depth over the next few years, it is expected that our methodology will enable simulations at a scale that lies beyond brute-force classical simulation, as we analyze in “Resource Scaling”, Section II F of our main text. We have also elaborated substantially on our analysis of quantum advantage in our response to comment #3 of Reviewer #3. Concretely, this **enables a route towards quantum advantage** as the problems we seek to tackle do “derive an advantage from these devices” (as described in *e.g.* [Nature 618, 500–505 (2023)])—this is intuitively how quantum advantage arises in comparison to classical methods.

With all our feats taken into consideration, we strongly believe that our manuscript meets the high standards of *Nature Communications* and is of interest to a large community working on quantum simulation of many-body systems.

Reviewer # 2

1. **Comment:** *The authors present how to map a single particle hopping on a d -dimensional lattice to d different particles hopping simultaneously on a 1-dimensional line. The original single particle creation (annihilation) operator in one certain dimension is replaced by the product of a specific creation (annihilation) operator of one kind with $(d-1)$ number operator of other kinds (representing rest dimensions). In other words, in the 1-dimensional line, each site now contains d qubits representing original dimensions. The authors then demonstrate impressively 2D, 3D, 4D robust topological corner modes on IBM hardware. Overall, I can recommend publication after the following issues are addressed.*

Response: We thank the reviewer for their positive evaluation of our work and the support for publication in *Nature Communications*. We are also glad that the reviewer recognizes the significance and value of our results, remarking that “the authors then demonstrate impressively 2D, 3D, 4D robust topological corner modes on IBM hardware”. Indeed our system is by far the largest effective condensed matter lattice model ever simulated on NISQ-era quantum hardware, and we believe that our work will pave the way for even more sophisticated quantum simulation as digital quantum computers improve in their error rates and qubit count in the near future. We address all their questions and comments herein.

2. **Comment:** *If my understanding is correct (first paragraph of this report), then Eq. 5 and Eqs. 7, 15 are not compatible. The non-hopping dimensions represented by $\prod_{\beta=1}^d n_{r_\beta}^\beta$ in Eq. 7 should be replaced with $\prod_{\beta=1, \beta \neq \alpha}^d n_{r_\beta}^\beta$, since there are $d-1$ (not d) degrees of freedom left.*

Response: We thank the reviewer for their careful reading and for pointing this out. Indeed, applying our mapping as specified in Eqs. (2) and (3) to our d -dimensional HOT lattice model in Eq. (6) yields a version of Eq. (7) that reads

$$\mathcal{H}_{\text{chain}}^{dD} = \sum_{\mathbf{r} \in [1, L]^d} \sum_{\alpha=1}^d u_{\mathbf{r}}^{\alpha} \left[(\omega_{r_{\alpha}+1}^{\alpha})^{\dagger} \omega_{r_{\alpha}}^{\alpha} \prod_{\substack{\beta=1 \\ \beta \neq \alpha}}^d n_{r_{\beta}}^{\beta} \right] + \text{h.c.} \quad (\text{R1})$$

But we remark that this is equivalent to the form of Eq. (7) given in our main text,

$$\mathcal{H}_{\text{chain}}^{dD} = \sum_{\mathbf{r} \in [1, L]^d} \sum_{\alpha=1}^d u_{\mathbf{r}}^{\alpha} \left[(\omega_{r_{\alpha}+1}^{\alpha})^{\dagger} \omega_{r_{\alpha}}^{\alpha} \prod_{\beta=1}^d n_{r_{\beta}}^{\beta} \right] + \text{h.c.} \quad (\text{R2})$$

To see this, we start from Eq. (R2) and pull out the $\beta = \alpha$ term, $n_{r_\alpha}^\alpha = (\omega_{r_\alpha}^\alpha)^\dagger \omega_{r_\alpha}^\alpha$, to get

$$\begin{aligned}
\mathcal{H}_{\text{chain}}^{dD} &= \sum_{\mathbf{r} \in [1, L]^d} \sum_{\alpha=1}^d u_{\mathbf{r}}^\alpha \left[(\omega_{r_{\alpha+1}}^\alpha)^\dagger \omega_{r_\alpha}^\alpha (\omega_{r_\alpha}^\alpha)^\dagger \omega_{r_\alpha}^\alpha \prod_{\substack{\beta=1 \\ \beta \neq \alpha}}^d n_{r_\beta}^\beta \right] + \text{h.c.} \\
&= \sum_{\mathbf{r} \in [1, L]^d} \sum_{\alpha=1}^d u_{\mathbf{r}}^\alpha \left[(\omega_{r_{\alpha+1}}^\alpha)^\dagger \omega_{r_\alpha}^\alpha \left[\mathbb{I} - \omega_{r_\alpha}^\alpha (\omega_{r_\alpha}^\alpha)^\dagger \right] \prod_{\substack{\beta=1 \\ \beta \neq \alpha}}^d n_{r_\beta}^\beta \right] + \text{h.c.} \quad (\text{R3}) \\
&= \sum_{\mathbf{r} \in [1, L]^d} \sum_{\alpha=1}^d u_{\mathbf{r}}^\alpha \left[(\omega_{r_{\alpha+1}}^\alpha)^\dagger \omega_{r_\alpha}^\alpha \prod_{\substack{\beta=1 \\ \beta \neq \alpha}}^d n_{r_\beta}^\beta \right] + \text{h.c.},
\end{aligned}$$

where we have used the fact that number operators commute and that $(\omega_{r_\alpha}^\alpha)^\dagger \omega_{r_\alpha}^\alpha + \omega_{r_\alpha}^\alpha (\omega_{r_\alpha}^\alpha)^\dagger = \mathbb{I}$ and $\omega_{r_\alpha}^\alpha \omega_{r_\alpha}^\alpha = 0$ for hardcore bosons. Thus the two versions of Eq. (7) are equivalent.

We have now changed the form of Eq. (7) to have the $\beta \neq \alpha$ labelling, as suggested by the reviewer. This should indeed be clearer to readers, and connects more directly to further technical workings in the quantum simulation methodology, for example Eqs. (15) and (16) in Methods.

3. **Comment:** *This new protocol seems plausible, but Eqs. 2, 3, 5 are given without proof. It is quite abstract and complicated at first sight, could the authors give some justifications? The proof can be intuitive or mathematical.*

Response: We thank the reviewer for their careful reading and their request for an expanded explanation behind our mapping formulation for quantum simulation.

Fundamentally, our core motivation is to sidestep limitations posed by the small number of available (high-quality) qubits on current-era NISQ hardware. The central idea is to map a d -dimensional lattice harboring a single particle into a one-dimensional chain harboring d particles, each of a different species and thus are distinguishable. We seek an exact mapping with no loss of information. For such a mapping to work, we need a way of encoding the position \mathbf{r} and sublattice γ of the particle in the original lattice into the one-dimensional chain system. We adopt the straightforward approach of identifying the location x of the α -th particle on the chain with the coordinate r_α of the original particle on the original lattice along the α -th spatial axis. The sublattice index γ is carried by every particle on the chain.

This mapping is exactly what Eq. (2) conveys. For clarity, we remind that $c_{\mathbf{r}\gamma}^\dagger, c_{\mathbf{r}\gamma}$ are operators for the original particle on d -dimensional lattice site \mathbf{r} and sublattice γ , and $[\omega_{x\gamma}^\alpha]^\dagger, \omega_{x\gamma}^\alpha$ are operators for the α -th particle on the chain, on location x and sublattice γ . In effect, the original particle is broken up into d distinguishable ones, each responsible for keeping track of the coordinate of the original particle along a particular spatial axis of the higher-dimensional lattice. Thus $c_{\mathbf{r}\gamma}^\dagger, c_{\mathbf{r}\gamma}$ are broken into products of $[\omega_{r_\alpha\gamma}^\alpha]^\dagger, \omega_{r_\alpha\gamma}^\alpha$ over the d different species, respectively, as written.

Eq. (3) is simply the generic lattice Hamiltonian with single-particle hoppings in Eq.(1), but with the mapping in Eq. (2) applied, which turns the $c_{\mathbf{r}\gamma}^\dagger, c_{\mathbf{r}'\gamma'}$ operators into products of $[\omega_{r_\alpha\gamma}^\alpha]^\dagger, \omega_{r'_\alpha\gamma'}^\alpha$. Physically, the hopping of the single particle on the original lattice is turned

into the simultaneous hopping of the d particles, such that the changes in the coordinates of the particle along the different spatial axes are respectively described. That is, the α -th particle on the chain hops to reflect changes in the α -th coordinate of the original particle on the lattice. Hopping of the original particle from sublattice γ to γ' on the lattice is reflected by the simultaneous hopping of all d particles on the chain from γ to γ' on the chain.

Eq. (4) is a 2D Hamiltonian harboring higher-order topology (HOT) on a square lattice, built from alternating SSH chains along both axes. This construction is known in literature—see for example Refs. [1,2]. Essentially, higher-order corner modes arise when SSH chains meeting at the corners are simultaneously in their topologically non-trivial regimes, such that their intersecting edge modes jointly produce corner localization. Broad theoretical analysis of this type of models is available in existing studies, and one generally thinks of them as higher-dimensional analogs of the standard SSH chain, but with higher-dimensional topological invariants (*i.e.* winding numbers) that represent higher-order bulk-boundary correspondence—*i.e.* with bulk topology determining the presence of boundary states one to a few co-dimensions lower. Note that in our writing of the Hamiltonian there is no sublattice degree of freedom—the SSH chains along each direction have different hoppings between even/odd sites—thus the sublattice index γ does not appear.

Lastly Eq. (5) is Eq. (4) with our mapping in Eq. (2) applied, or equivalently, specializing the generic Hamiltonian in Eq. (3) to the square lattice HOT Hamiltonian considered. To simplify notation and enhance clarity, we have written $\omega_\ell^1 = \mu_\ell$ and $\omega_\ell^2 = \nu_\ell$ for coordinate $\ell \in [1, L]$, as described in the text immediately above Eq. (5). We remind there is no need for a sublattice degree of freedom. We chose to present Eq. (5) explicitly as written, as it is one of the Hamiltonians we simulated in our experiments (as in Figure 3).

To return to the core motivation, the implication of this mapping procedure is that we have reduced the number of sites in the system from L^d in the original HOT lattice to L on the chain. This is at the cost of introducing simultaneous hoppings of all d particles on the chain, equivalently d -body interactions, to represent the original single-body hoppings on the original lattice. But as we utilize a quantum platform to perform quantum simulation, which has the versatility of accessing operations within the full Hilbert space of the qubit register, implementing these interactions is not particularly an issue—this is precisely the advantage of using quantum hardware. Only dL qubits are required to represent the chain, instead of L^d for a naive realization of the original lattice. The factor of d arises from our simplistic choice of using a group of d qubits to represent each site on the chain, such that the α -th particle is identified with the α -th qubit in the group.

A schematic of this mapping procedure, going from a 1-particle d -dimensional lattice of linear size L , to a d -particle 1-dimensional chain of L sites, and lastly to a dL -qubit chain, is schematically illustrated in Figure 2c. We have also added a new section to the Supplementary Information, Supplementary Note A, that describes the mapping formalism, alongside steps and considerations in derivations, in greater detail.

4. **Comment:** *The original model has 1 excitation in total, now the new model introduces d different particles, it's not clear to me how the new model conserves the total excitation.*

Response: We thank the reviewer for this question. The reviewer is absolutely correct that in applying the mapping to go from the original d -dimensional lattice model (with one excitation) to the one-dimensional chain (with d excitations), the number of excitations is not conserved—we refer back to our response to comment #3 for expanded details on the mapping formalism. This increase of the number of excitations, or “particles”, is exactly

what our mapping is intended to do: we map a 1-particle d -dimensional lattice of linear size L (*i.e.* with L^d sites) to a d -particle 1-dimensional chain of L sites, such that it can be encoded by dL -qubits. With fewer than d excitations on the chain, there cannot be sufficient degrees of freedom to exactly accommodate the original model as desired.

To clarify, in saying that $\mathcal{H}_{\text{chain}}^{dD}$ is number-conserving, we do not mean that the mapping is itself number-preserving: we mean that the Hamiltonian before and after the mapping are separately number-conserving. The number conservation for each of the d species of particles are symmetries of the chain Hamiltonian. That is, $[\mathcal{H}_{\text{chain}}^{dD}, n_{x\gamma}^\alpha] = 0$ where $n_{x\gamma}^\alpha = [\omega_{x\gamma}^\alpha]^\dagger \omega_{x\gamma}^\alpha$ is the number operator for the α -th species of particle on location x of the chain and sublattice γ , for every $x \in [1, L]$ and γ index (if sublattice structure is present). Equivalently $\mathcal{H}_{\text{chain}}^{dD}$ has d -fold U(1) symmetry.

This is the meaning intended, for example, in “the number-conserving symmetries of $\mathcal{H}_{\text{chain}}^{dD}$...” in Section II-B on page 5, and the referenced Methods subsections. As we explain in Methods, “Restricted sector”, this number-conservation allows a straightforward division of Fock space sectors of the chain into physical and unphysical partitions. Namely, as there is a single particle on the original lattice and as follows from our mapping, there is only a single particle of each species on the chain; number-conservation dictates that this remains true as time-evolution occurs. Hardware noise and inexact quantum circuits can cause departure from this physical sector. We can detect these departures in our measurements of site-resolved occupancies and discard experiment runs in which they happen. This is referred to as post-selection (see Methods, “Post-selection (PS)”) and is one of the error mitigation strategies we employ. Moreover, the number-conservation enables an optimization in our tensor-network aided circuit recompilation method—in particular, we can preferentially focus on the fidelity of the quantum circuit ansatz being optimized within physical sector of $\mathcal{H}_{\text{chain}}^{dD}$, instead of treating the entire Hilbert space equally. This is described in Methods, “Recompilation”.

We have made edits to the relevant parts of Section II B on page 5 and Methods that reference the number-conserving symmetries of $\mathcal{H}_{\text{chain}}^{dD}$ to improve clarity and to make the intended meaning more explicit.

5. **Comment:** *I’m not fully follow the explanation why there are $d \cdot 2^d$ hopping coefficients under Eq. 7. Now, d kinds of single particles have 2^d total states (occupied/ unoccupied for each species). Each site in the 1D chain comprises d qubits, the nearest-neighbor hopping is now mapped on to d qubits on site l interacting with d qubits on site $l + 1$. For all possible cases, there should be $2^d \cdot 2^d$ types of interaction. But since each particle has nearest neighbor hopping only with its own kind (hopping only occurs along one specific dimension), and conserves total excitation number, so interaction terms can be further reduced.*

Response: We thank the reviewer for the astute question. We agree completely that in a fully generic setting, there can be $2^d \cdot 2^d$ distinct hopping terms (from 2^d starting states to 2^d ending states). With complete generality for an arbitrary single-body Hamiltonian, as in Eq. (1), one thus has up to $2^d \cdot 2^d$ different hopping coefficients assuming sublattice degrees of freedom are absent, as the reviewer correctly points out.

Our counting of $d \cdot 2^d$ hopping coefficients applies in the more specific context of SSH-like HOT lattices, as we consider in our work from Eq. (4) onwards—even though our approach and mapping can be directly generalized to other types of lattices, for instance as elaborated in Supplementary Note A that we have added. While we start the discussion of the theoretical framework in the general setting of Eq. (1), we specialize to this class of HOT lattices for our quantum simulation demonstration and experiments on quantum hardware, which comprise

the bulk of the concrete results we showcase. This parameter count can be observed as follows. As noted in our response to comment #3 and in the main text (*e.g.* Section II A 2), our HOT lattices are built from alternating SSH chains along each axes. A standard SSH chain, following conventions in literature, has 2 distinct hopping coefficients, going from an odd to an even site (“intra-cell” hopping) and from an even to an odd site (“inter-cell” hopping). Thus, going along the axis of the chain, say in the direction of increasing coordinate, the hopping coefficients are labeled by the parity of the site x the hop originates from (of course one also adds the Hermitian conjugate at the end). Now, on the HOT lattice, which is the higher-dimensional analogue of the SSH chain, going along each of the $\alpha \in [d]$ axes, the hopping coefficients are labeled by the parities of the site \mathbf{r} the hop originates from, which is a vector in \mathbb{Z}_2^d . Thus the number of hopping coefficients is $d \times |\mathbb{Z}_2^d| = d \cdot 2^d$.

The counting argument described in Section II A 2 on page 4 is identical to the one above, but we have separated out the even/odd hopping coefficients along each direction, thus giving a tally of $d \times 2^{d-1} \times 2 = d \cdot 2^d$. We chose this separation as it is more consistent with the formalism in our paper, which denotes the odd/even coefficients along axis α as $v_{\pi(\mathbf{r})}^\alpha$ and $w_{\pi(\mathbf{r})}^\alpha$ respectively. We have added a footnote in page 4 of the main text to provide readers with the alternate approach discussed above.

6. **Comment:** *Although using less qubits, the authors’ protocol would require both long range and many-body interaction to create the required hopping between certain species. In section F, resources are counted in terms of Pauli-strings not the actual layers of two-qubits gates compiled on the hardware. This is not fair. I was wondering even after some tensor-network optimization how many layers of two-qubit gates used, say simulate the interactions appear in figure 3 (f)?*

Response: We thank the reviewer for the comment. The reviewer is correct that examining purely the number of Pauli strings is not a fair assessment. Indeed, we have therefore converted the Pauli string counts into estimates on the number of two-qubit gates required. We would like to refer the reviewer to our revised Section II F, page 11, where we describe the $\mathcal{O}(dL \cdot 2^{2d})$ Pauli string count per Trotter step, and thereafter we converted this number to $\mathcal{O}(d^2 L^2 \cdot 2^{2d}/\epsilon)$ total number of gates in the circuit in the worst-case for simulation precision of ϵ , assuming an all-to-all connectivity between qubits. This is relevant, for example, on trapped-ion and neutral-atom platforms (*e.g.* Quantinuum, QuEra) with ion/atom shuttling capabilities, which can bring arbitrary pairs of ions/atoms into suitable positions to facilitate two-qubit gate interactions (see *e.g.* Ref. [3])—but such movements generally incur decoherence noise. Imposing nearest-neighbor connectivity on the qubits, relevant for superconducting qubits (*e.g.* IBM, Google) and in general fixed-geometry platforms, does not change this $\mathcal{O}(d^2 L^2 \cdot 2^{2d}/\epsilon)$ bound.

Details of the Pauli string and gate counts are given in Methods, “Trotterization”. We have made edits to the referenced paragraph in Section II F, page 11 to emphasize the gate counts to readers.

The tensor-network aided quantum circuit optimization technique that we employed is hugely effective for quantum simulation circuits. Our group has invested considerable effort in developing this toolset—our prior works include *e.g.* Refs. [4–7]. As described in Methods, “Recompilation”, we did not find the need to exceed $K = 10$ layers of two-qubit gates in our optimized time-evolution circuits in our experiments, whilst maintaining fidelity exceeding 99.9% in the physical Fock space sector.

7. **Comment:** *There is a straightforward and well-structured way to map a d dimension lattice onto a 1d chain. For example, we can flatten the 2D lattice into 1D line (as shown below), the horizontal interactions are kept intact, but the vertical interactions now become the long range two-body interaction. I think at least for 2d case, this mapping may yield better result in terms of two-qubit gates than the authors.*

Response: We thank the reviewer for the suggestion and comment. Indeed, the “flattening” of a d -dimensional lattice onto a 1D chain by laying out segments end-to-end is commonly considered in literature. For example, it is the *de facto* standard way of performing classical MPS tensor-network simulation, which is inherently a 1D method, of 2D or higher-dimensional lattices.

We did not consider this conventional mapping suitable for our present work, as our core motivation was to sidestep qubit number limitations on NISQ devices, and this conventional mapping does not reduce the number of qubits needed—the motivation and intuition behind our approach has also been elaborated in our response to comment #3.

The “flattening” mapping referenced here preserves the number of sites in the system, preserves the single-particle nature of the system, and converts a portion of nearest-neighbour hoppings into long-range hoppings. A single qubit suffices to represent each site, and this qubit representation preserves the locality of the local hopping terms, mapping them onto spin operators on the qubits supporting those sites. But L^d qubits are then required to simulate the system, which is not better than the qubit requirements for a naive realization of the original higher-dimensional lattice (*i.e.* simply placing a qubit at each site of the lattice).

In contrast, the mapping used in our work reduces the number of sites to L , promotes the single-particle lattice into a d -particle chain, and converts all hoppings into potentially long-ranged multi-body interactions. As discussed, this reduces the qubit count requirement to only dL qubits, a drastic reduction from L^d for large L or d . This is at the cost of having to accommodate long-range interaction terms in our quantum circuits, which is exactly what the reviewer has correctly pointed out. Fortunately, the (digital) circuit implementation of long-range many-body interactions can well be executed on a gate-based platform (*e.g.* Ref. [8]). This advantage enabled by quantum hardware is a critical factor in why our work has been possible, and is positively encouraging with regard to its continued relevance.

Specifically regarding the number of two-qubit gates in the circuit, we agree with the reviewer that the referenced “flattening” mapping can result in a reduced gate count in some cases. However, we believe that there is a difference in scaling—and thus there exist regimes in which either approach has an advantage. In particular, using the “flattening” mapping, a nearest-neighbour hopping term on a d -dimensional cubic lattice can be turned into a $\mathcal{O}(L^{d-1})$ -ranged hopping, which requires $\mathcal{O}(L^{d-1})$ two-qubit gates to implement on a linear nearest-neighbour connectivity platform (which would be the fairest ground for comparison). Following the same counting argument as in our analysis (see *e.g.* Section II F or Methods, “Trotterization”), one has $d \cdot 2^d$ distinct hopping terms, thus giving $\mathcal{O}(d \cdot L^{d-1} \cdot 2^d / \epsilon)$ gates in the time-evolution circuit for simulation precision ϵ . This is in comparison to $\mathcal{O}(d^2 L^2 \cdot 2^{2d} / \epsilon)$ for our mapping. For large L and d , there is a regime in which our mapping can perform better, as we lack the L^d scaling—but for 2D lattices (or generically small d), the reviewer may be correct. Nonetheless, we clarify that our choice of mapping is motivated primarily by our desire to reduce qubit requirements, and only secondarily gate count, as discussed above.

8. **Comment:** *The results on hardware are quite impressive. I have some doubts about missing details. The total running time in Figures 3, 4, 5 is normalized to 1, but how does it compares*

to the qubit coherence time ?

Response: We thank the reviewer for the question. We clarify that in Figures 3, 4, and 5 of the main text, the time t is the simulation time for which the initial state undergoes Hamiltonian evolution over. That is, it is the time t in $|\psi_t\rangle = \exp(-i\mathcal{H}_{\text{chain}}^{\text{dD}} t) |\psi_0\rangle$ for an initial state $|\psi_0\rangle$ mapped onto the one-dimensional chain. On the original lattice, this is equivalent to $|\psi_t\rangle = \exp(-i\mathcal{H}_{\text{lattice}}^{\text{dD}} t) |\psi_0\rangle$ for single-particle $|\psi_0\rangle$ on the lattice.

The connection between t and the execution time τ of the quantum circuits implementing the unitary $\exp(-i\mathcal{H}_{\text{chain}}^{\text{dD}} t)$ on the quantum device is not precise, as the execution time τ depends sensitively on the structure of the quantum circuits and device specifications (*i.e.* gate times, readout times, and classical control capabilities). When trotterization is used, generally $\tau \propto t$. In order for the trotterization error to be controlled, the number of trotterization time steps must generically increase with t (such that each step covers a fixed time interval). The overall quantum circuit is the comprised of the time steps concatenated back-to-back (for example see Figure 2d of main text for illustration); thus circuit depth, consequently execution time, scales with t . The exact proportionality constant depends on the order of trotterization used and the specifics of the problem, such as the number and weight of Pauli strings in the Hamiltonian, and qubit connectivity and gate parallelization capabilities on the device.

This $\tau \propto t$ relation is muddied when circuit optimization techniques are applied. For example, under certain scenarios a constant-depth circuit can simulate time-evolution to arbitrarily large t [9]. For our Hamiltonians, which fall outside of certain “simpler” special classes, application of our tensor-network aided recompilation scheme does not guarantee that a fixed-depth circuit can reach arbitrarily large t ; but the dependence of circuit depth on t is weakened. For example, we did not find the need to exceed ~ 10 CX depth on our circuits after recompilation, which is at least two orders of magnitude shallower than naive trotterization.

At 10 recompilation ansatz layers, the circuit execution time is expected to be bounded below by $10\tau_{2q} + 11\tau_{1q} + \tau_{ro} \lesssim 10 \mu\text{s}$, with characteristic single-qubit gate time $\tau_{1q} \approx 40 \text{ ns}$, two-qubit gate time $\tau_{1q} \approx 400 \text{ ns}$, and readout time $\tau_{ro} \approx 4 \mu\text{s}$. Even accounting for initialization and control overheads, expected to be $\lesssim 4 \mu\text{s}$, the execution time is small compared to coherence times of $T_1, T_2 \approx 80 \mu\text{s}$ conservatively. In general, coherence times is not a primary limiting factor in our experiments; rather gate errors and hardware noise are.

9. **Comment:** *In the caption of Figures 3, 4, 5, symbols like v_0^x are not defined. The definitions only appear in the very end of the paper.*

Response: We thank the reviewer for this crucial feedback. The symbols $v_{\pi(\mathbf{r}_\alpha)}^\alpha$ and $w_{\pi(\mathbf{r}_\alpha)}^\alpha$ are intra- and inter-cell hopping coefficients respectively, along the α -th direction and labeled by the parity of site coordinates \mathbf{r}_α excluding the α -th coordinate. This is described in Section II A 2, page 4, in the text above Eq. (7). This notation is a generalization of the more specific notation in for instance Eq. (4), to accommodate an arbitrary number of dimensions d .

For example, v_0^x and w_0^x appearing in Figure 3 are intra- and inter-cell hopping coefficients with $\alpha = x$ and $\mathbf{r}_\alpha = 0$, that is, hopping coefficients along the x direction, for starting sites of odd y coordinate. To be explicit, we note that throughout our work, the index α is used to enumerate the spatial axes and also the particle species on the 1D chain, which are equivalent under our mapping—hence α takes on values x, y, z, \dots and $1, 2, 3, \dots$ synonymously. Likewise, v_1^x and w_1^x are intra- and inter-cell hopping coefficients along the x direction, for starting sites of even y coordinate.

We have made edits to the defining sentence in Section II A 2, page 4 to emphasize the meaning of $v_{\pi(\mathbf{r}_\alpha)}^\alpha$ and $w_{\pi(\mathbf{r}_\alpha)}^\alpha$. We have also edited our captions of Figures 3, 4 and 5 to more clearly refer to v_{π}^α and w_{π}^α as the hopping coefficients on the lattice, which are enumerated by axis α and parity π .

10. **Comment:** *I suggest replace RO (readout Error Mitigation) with EM, it took a while to search for this unfamiliar term.*

Response: We thank the reviewer for the kind suggestion. We clarify that the stack of error mitigation techniques employed in our experiments comprises not only readout error mitigation, which seeks to approximately correct for bit-flip errors during measurements, but also post-selection, which detects and discards results in unphysical sectors that may arise due to hardware noise and gate miscalibrations. To distinguish between these, we have denoted them with abbreviations RO and PS respectively in our figures. We respectfully decline to rename RO to EM (which we presume stands for error mitigation), as we are worried that it is insufficiently descriptive and may cause readers to unduly infer that readout error mitigation is the only strategy utilized in our work.

We apologize for the difficulty in searching for the abbreviation. To make reading easier, we have gone over the manuscript and ensured that references to the abbreviations (RO, PS) are placed in brackets throughout the text, such as in the caption of Figure 2, Section II B on page 5, and in Methods. We note that these abbreviations have been used in prior and recent works, for example Refs. [4, 5, 10–12].

11. **Comment:** *In the supplementary table, it is not clear how the parameters (v_0^x etc.) are mapped to the SSH-like model*

Response: We thank the reviewer for pointing this out, and we apologize for the lack of clarity. To understand how the parameters are related to the SSH-like HOT lattice models, we politely refer the reviewer to our response to comment #9, where we have elaborated on the notation and the physical meaning of v_{π}^α and w_{π}^α symbols. In particular, they are intra- and inter-cell hopping coefficients on the lattice labeled by the hopping direction $\alpha \in \{x, y, z, \dots\}$ and the parity of the lattice site coordinates in the remaining $d - 1$ dimensions, $\pi \in \mathbb{Z}_2^{d-1}$.

To improve clarity, we have edited the caption of Supplementary Table 1 to explain explicitly the meaning of v_{π}^α and w_{π}^α , and to draw the connection to the Hamiltonian and definitions in the main text. This accompanies edits in Section II A 2, page 4 and captions of Figures 3, 4 and 5 that are also to improve clarity on the notation.

Reviewer # 3

1. **Comment:** *In manuscript NCOMMS-23-43592-T, the authors proposed to simulate multi-dimensional higher-order topological models on quantum hardware. The key point lies in the sufficient reduction of lattice sizes by mapping single-particle high-dimensional lattices to one-dimensional interacting qubit chains. Compared with previous quantum simulations through using synthetic dimensions, they realized higher-order topological states in real-space high-dimensional lattices. The results are clear and concrete. To the best of our knowledge, it is the first experimental realization of 4D higher-order topological states in quantum systems.*

Response: We thank the reviewer for their careful reading and evaluation of our work.

Indeed, our present study reports the first experimental realization of a genuine 4D higher-order topological (HOT) states in a purely quantum setting; and unlike prior works, our work focuses on realizing real-space high-dimensional lattices without invoking synthetic dimensions, which is crucial for realizing boundary phenomena such as topological corner modes (and not just band structures in parameter space). The innovations we present, such as the mapping scheme mentioned and our quantum simulation methodology, including circuit construction and methods to address noise in experiments, go towards overcoming challenges associated with these goals, and also paves the path for future experimental realization of high-dimensional interacting quantum systems for which alternative experimental platforms are challenging to design.

2. **Comment:** *However, it is unclear whether they can use their mapping scheme to study more complex structures such as Kagome lattice which can also support higher-order corner states [Nature Materials 18, 108 (2019), Nature Photonics 13, 697 (2019)]. If their scheme is only applicable to simple square, cubic, tesseract lattices, this work may lack broad research interest. Furthermore, even without performing experiments, constructing higher-order topological corner states from one-dimensional SSH-type model is well known [PRL 118, 076803 (2017), PRL 122, 233902 (2019), PRB 104, 224303 (2021)]. These results are quite natural and expected and of less novelty.*

Response: We thank the reviewer for this important concern on the generality of our approach. We would like to clarify that our mapping scheme is fully general, and readily adaptable to any lattice geometry hosting arbitrary hopping processes and on-site potentials, including triangular, Kagome, and BHH lattices—we refer the reviewer to our later response to comment #4 for more details. As a concrete response here, we remark that the way our mapping has been formalized, as in Section II F, deliberately does not assume any underlying lattice geometry. To accommodate a lattice and model of interest, one specifies the sites $\{\mathbf{r}\}$ and sublattices $\{\gamma\}$, and the hopping coefficients $\{h_{\mathbf{r}\mathbf{r}'}^{\gamma\gamma'}\}$ that define the model directly carries over onto the mapped quantum chain. The sites present on the quantum chain is the union of distinct site coordinates on the lattice along each direction. We elaborate on the structure and usage of the map in Supplementary Note A 1 which we have added, alongside concrete examples of triangular and Kagome lattices, and the BBH model, that we have worked out and provided technical elaborations on.

In fact, our mapping is also applicable to lattices hosting multiple interacting particles, such as Hubbard or fractional quantum-Hall-type interactions, as we also describe in our response, although the study of many-body physics is beyond the scope of our present experiments and is currently the subject of subsequent investigations in our group. We elaborate on this in Supplementary Note A 2.

We agree with the reviewer that the construction of lattices hosting higher-order topological corner modes from SSH-type chains as building blocks is known in the literature; the lattice Hamiltonians we consider are in fact adapted from prior studies (*e.g.* [1, 2]) as referenced. Though, we remark that this is the first time a 4D HOT lattice model has been explicitly studied, to the best of our knowledge, without the use of synthetic dimensions. Nonetheless, we would like to emphasize that the main focus and contribution of our work is not in the construction nor the theoretical analysis of these SSH-type HOT models; rather it is in the direct experimental realization of these higher-dimensional lattices on superconducting quantum hardware, and in developing and demonstrating the pipeline of methods required for such a task. Successfully performing the quantum simulation on hardware is non-trivial and can by no means be taken for granted, considering the serious constraints (*i.e.* qubit numbers, qubit connectivity, gate error rates, decoherence, *etc.*) on current-era experiment platforms; the results presented would be impossible without the advances in methodology we describe.

3. **Comment:** *The mapping scheme can reduce the number of qubits from L^d to dL , which significantly save resources of quantum devices. However, for a single-particle problem, the dimension L^d is accessible in classical computer. The quantum advantages are not obvious in dealing with single-particle higher-order topological problem. Actually, to demonstrate corner states, the size $L = 10$ is enough. One can immediately obtain $L^4 = 10000$ eigenstates, eigenvalues and the dynamics on laptop computer.*

Response: We thank the reviewer for their astute comment. We agree that an $L = 10$, $d = 4$ lattice harboring a single-particle Hilbert space of dimension $L^4 = 10000$ dimensions is readily accessible on a laptop computer. Indeed the exact diagonalization (ED) results shown for comparison against datapoints obtained on the quantum processors were computed classically, and we correspondingly make no claims that any of the experiment results presented are “beyond-classical”, so to speak.

The resource advantage of our quantum simulation protocol discussed in Section II F of the main text should be understood as manifesting in an asymptotic sense—*i.e.* as one increases the linear lattice size L or dimension d to a sufficient degree. Unfortunately, due to noise and qubit number limitations on present-day quantum processors, we were not able to obtain conclusive results for much larger L . But given the rapidly improving quantum hardware, we are positive that this resource advantage can be experimentally realized in the next 3–5 years. For example, IBM has recently reported a larger 133-qubit device with \sim halved two-qubit error rates and virtually no crosstalk error, in comparison to the predecessor processor family.

This theoretical resource advantage should be clear from the difference in the growth rates of execution time $\mathcal{O}(L^{3d})$ for generic classical dynamics simulation and $\mathcal{O}(d^2 L^2 \cdot 2^{2d}/\epsilon)$ for quantum simulation, the latter circumventing exponentials of L in the dimension of the lattice. A crossover occurs for sufficiently large L or d beyond which quantum advantage manifests. The precise boundary at which this crossover occurs depends on platform-specific details such as gate times and control overheads, which affect the constant factors in the resource scalings. We discuss the resource requirements and the crossover boundary for quantum advantage in more detail in our response to comment #5.

We additionally clarify that while the experiment results in our paper pertain to single-particle HOT lattices, which was our subject of interest, our mapping and quantum simulation methodology is applicable also to interacting m -particle systems ($m > 1$) with no fundamental modifications, only at the expense of a larger local Hilbert space dimension (to accommodate

the m particles) and more complicated interactions on the chain—we point this out to readers in our Discussion and outlook, and we have now also added Supplementary Note A, which provides a detailed accounting of the mapping in both single- and multi-particle contexts. This generality is a natural consequence of the structure of our mapping, which functions at the level of particle creation/annihilation operators on the lattice and thus equally well treats multi-body interaction terms.

Allow us to comment further on assessing resource requirements in terms of scaling with respect to problem parameters, as was done in our work. The well-known statements that quantum advantage is realizable in, for instance, factoring or period-finding problems (*i.e.* Shor’s algorithm), unstructured search (*i.e.* Grover’s algorithm and amplitude amplification), solving linear systems (*i.e.* HHL algorithm), precision measurements in quantum metrology (*i.e.* squeezing or collective measurement protocols), and optimization problems (*i.e.* annealing, quantum walks) are of all this nature—see, for example, excellent surveys in Refs. [13,14]. As gate times and overheads in quantum processors are orders of magnitude slower than their comparable analogues in classical computers, at least with foreseeable technology, one does not generally expect quantum computation to yield a practical advantage for small problem sizes regardless of the problem—for instance factoring a $\mathcal{O}(10)$ -bit number is best done on a laptop but a $\mathcal{O}(2000)$ -bit cryptographically relevant counterpart would be more quickly treated on a quantum processor. The simulation problem considered in our work is no different in this respect.

As a concluding remark, we take a broader higher-level perspective and provide contextualization of the value provided by our work to the community. A recent perspective on quantum many-body simulations [Nature Communications 15, 2123 (2024)] pointed out that “while some many-body interactions are not physically realizable in analog quantum simulators, their digital decomposition can very well be implemented on a gate-based platform”. Our current work is the first to concretely demonstrate the digital quantum simulation of quantum systems of comparable complexity and scale as our d -dimensional HQT lattices on current-day hardware, and to provide a scalable route for sustained advancement of the simulation of this type of models as hardware improves. Our results carry far-reaching implications because, foundationally, we fully utilize the “quantum-ness” of the simulation platform to our advantage, namely the many-body Hilbert space and the many-body interactions the platform supports. Thus the problems we seek to tackle do “derive an advantage from these devices” (as described in [Nature 618, 500–505 (2023)]), an intuitive requisite for quantum advantage to arise. As quantum hardware improves in error rates and attainable circuit depth over the next few years, our methodology eventually becomes able to perform simulations at a scale that lies beyond brute-force classical simulation.

4. **Comment:** *The authors should comment whether their scheme can apply to general lattice such as triangle lattice and Kagome lattice. Besides, since the results of two-dimensional SSH model are well known, the authors should properly cite existing references, and explicitly show how to apply their mapping scheme to a more complex lattice such as Kagome lattice or Benalcazar-Bernevig-Hughes (BBH) lattice.*

Response: We thank the reviewer for the suggestion and comment. The generality of our scheme is indeed an important consideration, particularly since models of interest to the condensed matter community commonly involve triangular and Kagome lattices, which the reviewer has kindly pointed out. We assure the reviewer that our mapping scheme is fully general and is applicable to any lattice geometry, hosting arbitrary hopping configurations

and on-site potentials. In Supplementary Note A 1 that we have added, we provided an extended technical discussion of the general mapping procedure, and demonstrate explicitly the application of our mapping to triangular, Kagome, and BBH lattices; from the formalism and examples, it should be evident that our approach would likewise work for many other generic models, up to minor modifications as one may desire. Additionally, in Supplementary Note A 2, we provide elaboration on how generic multiple-particle interactions on the lattice can also be implemented in our approach.

We hope that these new materials would convince the reviewer that our approach is indeed broadly applicable and versatile for realizing a wide variety of both non-interacting and interacting physics. Below, we explain the ideas behind its generality.

The generality of the mapping follows from its starting point (at the end of page 2, Section II A 1 of the main text), where we consider the most general d -dimensional n -band model in second-quantized form. Here, the specific lattice structure and Hamiltonian terms are encoded in the Bloch Hamiltonian $\mathcal{H}(\mathbf{k})$. In Eq. (1) we express this Hamiltonian in real space, retaining full generality in the lattice geometry, encoded by site coordinates $\{\mathbf{r}\}$ and sublattices $\{\gamma\}$, and the hoppings $\{h_{\mathbf{r}\mathbf{r}'}^{\gamma\gamma'}\}$ between sites and sublattices. The specification of our mapping is independent of these details; one simply follows the recipe that we lay out on the Hamiltonian of interest (setting $\{\mathbf{r}\}$, $\{\gamma\}$ and $\{h_{\mathbf{r}\mathbf{r}'}^{\gamma\gamma'}\}$ to specify the model). In our response to comment #3 of Reviewer #2, we have provided an expanded discussion of the motivation and considerations underlying the mapping scheme, which to avoid repetition, we do not reproduce here. We kindly refer the reviewer to that response for background context.

Since our mapping functions at the level of particle creation/annihilation operators— $c_{\mathbf{r}\gamma}^\dagger, c_{\mathbf{r}\gamma}$ for arbitrary sites $\{\mathbf{r}\}$ and sublattice structure $\{\gamma\}$ on the lattice—we can accommodate any starting Hamiltonian given in second-quantized form. In particular, the mapping proceeds by representing the original single particle on the d -dimensional lattice with d distinguishable particles on a one-dimensional chain, each a different species. The α -th particle is responsible for encoding (via its location on the chain) the coordinate of the original particle along the α -th axis of the lattice; the sublattice label is carried by all particles. Thus $c_{\mathbf{r}\gamma}$ is replaced by a product of $\omega_{r_\alpha\gamma}^\alpha$ operators over $\alpha = 1, 2, \dots, d$, and likewise for $c_{\mathbf{r}\gamma}^\dagger$. Essentially, the single-particle lattice Hamiltonian is turned into a d -particle chain Hamiltonian, with single-body hoppings transformed into simultaneous hoppings of the d particles, or equivalently, d -body interaction terms.

This replacement of $c_{\mathbf{r}\gamma}$ with $\omega_{r_\alpha\gamma}^\alpha$ is independent of lattice structure, and is straightforwardly applicable to other lattices with a generic number of sublattices/bands, beyond what we have considered in our experiments. Different models on different lattices are specified by $\{\mathbf{r}\}$, $\{\gamma\}$ and $\{h_{\mathbf{r}\mathbf{r}'}^{\gamma\gamma'}\}$; on the quantum chain, coordinates $\{x\}$ follow from $\{\mathbf{r}\}$, namely comprising the union of $\{r_\alpha\}$ coordinates over each direction α , the sublattice structure $\{\gamma\}$ is copied as-is, and $\{h_{\mathbf{r}\mathbf{r}'}^{\gamma\gamma'}\}$ translates as the interaction amplitudes on the chain. Once the chain Hamiltonian is written, the remainder of our quantum simulation methodology goes through unchanged—no stage of our circuit construction, optimization, execution, and error mitigation methodology is dependent on the specific structure of the lattice Hamiltonian.

As mentioned, we have now added Supplementary Note A, which provides a detailed walk-through of our mapping scheme. In Supplementary Note A 1, we demonstrate explicitly the application of our mapping to triangle, Kagome, and BBH lattices as requested. Additionally, in Supplementary Note A 2, we provide explicit details on the natural generalization of our mapping for lattices hosting multiple interacting particles.

5. **Comment:** *Fig. 7 should give values of both x and y axis, so that we can know what are the transition points when the mapping scheme with quantum computer is better than classical exact diagonalization.*

Response: We thank the reviewer for the feedback. Indeed, in analyzing the difference in resource requirements between our quantum simulation approach and classical numerical methods, obtaining estimates of the crossover point leading into quantum advantage can be helpful.

To start, we must emphasize that the exact transition boundary depends sensitively on details specific to the quantum platform used, namely, on gate times and gate parallelization capabilities. For instance, superconducting qubits have intrinsic energy scales (*i.e.* gap between $|0\rangle$, $|1\rangle$ computational states and qubit interaction strengths) in the 10 MHz–1 GHz range, and consequently time scales on the order of 1–100 ns, with inherent capability to execute gates acting on disjoint qubits in a fully parallel fashion as each qubit (and couplers between qubits when present) typically have dedicated control lines. To be concrete, the IBM devices used in our experiments have single-qubit gate times of ~ 40 ns or shorter, two-qubit CX gate times of ~ 400 ns, and measurement times of ~ 800 ns or longer; other hardware designs, such as the chips by Google, can perform (non-CX) entangling gates in ~ 50 ns or less. Photonic and silicon-based platforms theoretically share similar timescales, though technologically are not as mature and no processor of comparable scale have been reported to date. Trapped ion and neutral atom platforms have inherently longer gate times, on the order of 5–200 μ s, though recent progress have pushed gate times to sub-1 μ s levels; for entangling gates facilitated by Rydberg blockade, for example, a general path forwards is to increase interaction energies by bringing the atoms closer together with stronger trapping lasers. Fully parallel execution of gates on trapped ion platforms have been reportedly difficult due to crosstalk and control difficulties, and typically neutral atom arrays utilize global controls (instead of local)—though this architectural choice also simplifies operation and benefits scalability.

Regardless, in the current landscape, one observes a difference in gate timescales differing by ~ 3 orders of magnitude or more, as well as differences in parallelization capabilities, which drastically affect the scheduling of operations. It is also difficult to predict the changes in these specifications in the future, given the rapid advancement and evolution in quantum platforms. As an illustration to the volatility of device metrics, IBM reduced measurement times by a factor of ~ 8 essentially in a span of a year; and robust shuttling of large neutral atom arrays with mid-circuit measurement capability have only been recently demonstrated (see *e.g.* [15]).

We believe, therefore, that it is not meaningful to produce a definite numerical estimate for the crossover point, presumably specialized to some set of assumed device specifications—any such projection is almost surely doomed to be inaccurate by wide margins. We have accordingly chosen not to include specific numbers in our Figure 7 plots, which may be misleading to readers or inaccurate in the very near future. The objective of our work is to concretely demonstrate the quantum simulation of HQT lattices, and we bring to attention that the mapping and quantum simulation methodology developed provides an asymptotic resource advantage—this statement is true, *i.e.* quantum advantage is realizable, regardless of platform.

Nonetheless, to provide a clearer picture, here we complement Figure 7 with three rough scenarios—a “neutral” projection, a pessimistic, and an optimistic counterpart—as described below. In all scenarios we consider superconducting qubit platforms; the timescales differ

greatly for other platforms, as discussed above. We take a simulation precision $\epsilon = 0.1\%$ for the quantum simulation approach in all scenarios. For classical exact diagonalization, we assume a processor throughput of 1×10^{12} floating-point operations per second, which is in the ballpark of performance of the best consumer CPUs on the market (*e.g.* AMD 5950X). We assume $d^2 L^2 \cdot 2^{2d} / \epsilon$ layers of gates for quantum simulation and L^{3d} floating-point operations for exact diagonalization, ignoring constant factors that are potentially present (the effect of constant multipliers on the crossover point is small considering the exponentials).

- *As-is scenario.* Here we assume that gate times remain similar to present-day values but qubit coherence have been improved, or quantum error correction applied without significant overhead, such that arbitrarily deep circuits can be executed without concern. In particular we adopt a timescale of 400 ns per gate layer.

Figure R1: Variant of runtime scalings calibrated for the as-is scenario.

- *Pessimistic scenario.* Here we assume that quantum error correction has introduced significant slowdowns in the quantum processor operation rate, in comparison to the “raw” physical operations as we have now. This is because in each round of quantum error correction, operations must be performed between logical data qubits and ancillae, the ancillae measured, and the syndromes feedforward into correction operations—this sequence naturally requires more time than a bare operation on physical qubits. Generally quantum error correction rounds are repeated frequently, on the order of 10 MHz, interleaved with logical operations of the circuit for fault-tolerant quantum computation. We take as ballpark a 100-fold slowdown in effective gate time, thus a timescale of 40 μ s per gate layer.

Figure R2: Variant of runtime scalings calibrated for the pessimistic scenario.

- *Optimistic scenario.* Here we assume that advancements in hardware and control techniques are such that, despite quantum error correction overheads discussed above, the effective gate times are reduced. We take as ballpark 10 ns per gate layer.

Figure R3: Variant of runtime scalings calibrated for the optimistic scenario.

6. **Comment:** *In Sec.II C, the authors declare “Previously, such HOT corner modes on 2D lattices have been realized in various metamaterials [34, 35]...”. It should be noted that the topological corner states in two-dimensional lattices have been experimentally observed in photonic waveguides [PRB 105, 195129 (2022)].*

Response: We thank the reviewer for the comment. We have added the reference to the mentioned paper in Section II C.

7. **Comment:** *In Sec. II C, the authors claimed that “Our equivalent 1D hardcore boson chain is also the first demonstration of interaction-induced higher-order topology...”. We do not think it is interaction-induced higher-order topology. The authors used one dimensional interacting boson chain to simulate two dimensional non-interacting higher-order topological lattice. In one dimensional systems, there is no higher-order corner states at all. They can make such claim only when the original two dimensional lattice is trivial and interaction between particles in two dimensions gives rise to the higher-order topological states.*

Response: We thank the reviewer for their comment. We agree that this claim of “interaction-induced higher-order topology” is misleading, and we agree with the perspective of the reviewer in interpreting the meaning of the term, which is certainly consistent with the standard understandings of interaction-induced topology in the field. We have thus removed our statement about “interaction-induced higher-order topology”. To be clear to readers, we have made edits in the Introduction and Section II C, pointing out that the topology of the HOT lattice is encoded on the joint degrees of freedom (*i.e.* joint configuration space) on the quantum chain, which is purely interacting.

8. **Comment:** *Ranging from two-dimensional square lattice and three-dimensional cubic lattice to four-dimensional tesseract hyperlattice, the direct measurement of dynamical evolution and energy spectrum serves as evidences for higher-order topological states. Not limited to the observation of topological boundary states, it is better to comment how to experimentally measure the topological invariant on quantum hardware. This is important for bulk-boundary correspondence as you mentioned in the forth paragraph of the introduction.*

Response: We thank the reviewer for their suggestion. The reviewer is certainly correct in stating that a direct measurement of the topological invariant associated to the higher-order topology would be the smoking-gun evidence for HOT states. Conventionally, to perform such a task, the topology of the wavefunction has to be inferred via elaborate quantum circuits that seek to measure the local wavefunction overlap with its neighbouring wavefunction in parameter space, *i.e.* holonomy. This has been recently attempted on hardware for simple interacting and non-interacting models [16], via the use of a variational quantum eigensolver

(VQE)-based approach. Variational schemes are known to be vulnerable to hardware noise (as they involve classical-quantum optimization dependent on measured data) and face a number of theoretical problems, such as barren plateaus, that worsen with system size; thus, while the results are definitely impressive, we believe the pathway forward for larger more complex systems (*e.g.* the higher-dimension HOT lattices examined here) is unclear, and likely cannot be a straightforward scaling-up of the methods demonstrated thus far.

Since measuring the topological invariant for generic HOT lattices is not a research goal that we seek to tackle, we chose to probe the topology of our lattices by measuring the number of edge states (zero/low energy spectrum). By bulk-boundary correspondence, this is in fact equivalent to probing the topological invariant. For instance, the number of HOT modes exactly correspond to the topological invariant (*i.e.* higher-dimensional winding number) in this class of lattices; and as we have realized the system in real space and not momentum space, the direct counting of the edge states is more practical. In our study, we observed the presence of corner-, edge-, and surface-localized robust HOT modes in our time-dynamics experiments, and we also measured the zero-energy topological spectrum of the 2D and 3D lattices via IQPE, as shown in the insets of Figures 3 and 4, which allows a direct survey of the number of HOT modes.

We have discussed the challenges and difficulty in successfully performing high-fidelity time-evolution simulations of the lattices in our previous responses—but here we remark also that our IQPE results are likewise highly non-trivial, as quantum phase estimation is widely considered a demanding task on quantum hardware, due to the large circuit depths and numbers of gates needed. Our choice of an iterative version of quantum phase estimation, which trades requirements on quantum circuit breadth and depth for a larger number of distinct circuits run sequentially (but each of lower breadth and depth), in combination with our mapping, error mitigation and circuit optimization techniques, allowed us to yield good measurements of topological midgap spectrum on noisy presently-available hardware.

Thus, we believe that not directly measuring the topological invariant on quantum hardware does not reduce the validity of our investigation higher-order topology or detract from the value of our work—though we agree completely that further work in this direction is valuable and important.

Reviewer # 4

1. **Comment:** *I co-reviewed this manuscript with one of the reviewers who provided the listed reports. This is part of the Nature Communications initiative to facilitate training in peer review and to provide appropriate recognition for Early Career Researchers who co-review manuscripts.*

Response: We are grateful for the reviewer's joint suggestions, and we sincerely thank the reviewer for their time.

References

- [1] Benalcazar, W. A., Bernevig, B. A. & Hughes, T. L. Quantized electric multipole insulators. *Science* **357**, 61–66 (2017).
- [2] Li, L., Umer, M. & Gong, J. Direct prediction of corner state configurations from edge winding numbers in two- and three-dimensional chiral-symmetric lattice systems. *Phys. Rev. B* **98**, 205422 (2018). URL <https://link.aps.org/doi/10.1103/PhysRevB.98.205422>.
- [3] Palani, D. *et al.* High-fidelity transport of trapped-ion qubits in a multilayer array. *Phys. Rev. A* **107**, L050601 (2023). URL <https://link.aps.org/doi/10.1103/PhysRevA.107.L050601>.
- [4] Koh, J. M., Tai, T., Phee, Y. H., Ng, W. E. & Lee, C. H. Stabilizing multiple topological fermions on a quantum computer. *npj Quantum Inf.* **8**, 16 (2022).
- [5] Koh, J. M., Tai, T. & Lee, C. H. Simulation of interaction-induced chiral topological dynamics on a digital quantum computer. *Phys. Rev. Lett.* **129**, 140502 (2022). URL <https://link.aps.org/doi/10.1103/PhysRevLett.129.140502>.
- [6] Shen, R., Chen, T., Yang, B. & Lee, C. H. Observation of the non-hermitian skin effect and fermi skin on a digital quantum computer. *arXiv preprint arXiv:2311.10143* (2023).
- [7] Chen, T., Shen, R., Lee, C. H. & Yang, B. High-fidelity realization of the aklt state on a nisq-era quantum processor. *SciPost Physics* **15**, 170 (2023).
- [8] Fauseweh, B. Quantum many-body simulations on digital quantum computers: State-of-the-art and future challenges. *Nature Communications* **15**, 2123 (2024).
- [9] Bassman Oftelie, L. *et al.* Constant-depth circuits for dynamic simulations of materials on quantum computers. *Materials Theory* **6**, 13 (2022).
- [10] Majumdar, R., Rivero, P., Metz, F., Hasan, A. & Wang, D. S. Best practices for quantum error mitigation with digital zero-noise extrapolation. In *2023 IEEE International Conference on Quantum Computing and Engineering (QCE)*, vol. 1, 881–887 (IEEE, 2023).
- [11] Koh, J. M., Sun, S.-N., Motta, M. & Minnich, A. J. Measurement-induced entanglement phase transition on a superconducting quantum processor with mid-circuit readout. *Nat. Phys.* (2023).
- [12] Reinhold, P. *et al.* Error-corrected gates on an encoded qubit. *Nature Physics* **16**, 822–826 (2020).
- [13] Dalzell, A. M. *et al.* Quantum algorithms: A survey of applications and end-to-end complexities (2023). [2310.03011](https://arxiv.org/abs/2310.03011).
- [14] Montanaro, A. Quantum algorithms: an overview. *npj Quantum Information* **2**, 1–8 (2016).
- [15] Bluvstein, D. *et al.* Logical quantum processor based on reconfigurable atom arrays. *Nature* **626**, 58–65 (2024).
- [16] Xiao, X., Freericks, J. K. & Kemper, A. Robust measurement of wave function topology on nisq quantum computers. *Quantum* **7**, 987 (2023).

REVIEWERS' COMMENTS

Reviewer #1 (Remarks to the Author):

I have a look of the responses to reviewing reports and the revised version of the paper, now, I agree that the realization of four dimension higher-order topological state is the first simulation of such a topological phenomenon. In general, the resources needed in simulation are comparable with the number of qubits in a lattice model. The paper shows that the resources scale with system size and dimension of the model, which demonstrate the power of quantum simulation and may solve some complicated computational problem. This quantum simulation work is interesting, I think that it can be published in Nature Communications.

Reviewer #2 (Remarks to the Author):

My comments are addressed, I recommend the publication of this manuscript.

Reviewer #3 (Remarks to the Author):

In the second version of the manuscript the authors present an improved version. Answers to all referee questions have been provided. The authors propose a general and exact method to map a high-dimensional lattice to a one-dimensional chain with non-local many-body interactions. This scheme provides a powerful platform to simulate important physics in higher-order topology and high-dimensional lattices, which overcomes the key challenges of gate errors, quantum decoherence and qubit number limitation. Given that the authors address novel aspects relevant to quantum simulation, the contribution can be considered significant and is expected to generate broad interest for readers. The authors' statements are supported by a checkable mathematical derivation, and this provides grounds for judging the work as an authoritative and substantive addition to the body of literature. Finally, the subject matter is explored comprehensively.

Based on above, We conclude that all the formal requirements have been fulfilled and the manuscript could be accepted for publication.

Reviewer #4 (Remarks to the Author):
